# Meflin-positive cancer-associated fibroblasts enhance tumor response to immune checkpoint blockade

Yuki Miyai[1,2], Daisuke Sugiyama[3], Tetsunari Hase[4], Naoya Asai[1], Tetsuro Taki[1], Kazuki Nishida[5], Takayuki Fukui[6], Toyofumi Fengshi Chen-Yoshikawa[6], Hiroki Kobayashi[1], Shinji Mii[1], Yukihiro Shiraki[1], Yoshinori Hasegawa[4], Hiroyoshi Nishikawa[3,7], Yuichi Ando[2], Masahide Takahashi[1], Atsushi Enomoto[1]

Cancer-associated fibroblasts (CAFs) are an integral component of the tumor microenvironment (TME). Most CAFs shape the TME toward an immunosuppressive milieu and attenuate the efficacy of immune checkpoint blockade (ICB) therapy. However, the detailed mechanism of how heterogeneous CAFs regulate tumor response to ICB therapy has not been defined. Here, we show that a recently defined CAF subset characterized by the expression of Meflin, a glycosylphosphatidylinositol-anchored protein marker of mesenchymal stromal/stem cells, is associated with survival and favorable therapeutic response to ICB monotherapy in patients with non-small cell lung cancer (NSCLC). The prevalence of Meflin-positive CAFs was positively correlated with CD4-positive T-cell infiltration and vascularization within non-small cell lung cancer tumors. Meflin deficiency and CAF-specific Meflin overexpression resulted in defective and enhanced ICB therapy responses in syngeneic tumors in mice, respectively. These findings suggest the presence of a CAF subset that promotes ICB therapy efficacy, which adds to our understanding of CAF functions and heterogeneity.

## Introduction

As immune checkpoint blockade (ICB) therapy is emerging as a promising treatment strategy for a wide range of cancers, understanding the mechanisms underlying tumor immunity and identifying biomarkers that predict patient outcomes has been a focus of cancer research (Rizvi et al, 2015; Kumagai et al, 2020a, 2020b; House et al, 2020; Mager et al, 2020; Smith et al, 2021). The intrinsic

properties of tumor cells, mutational burdens, and their interactions with host immune cells are critical for the efficacy of ICB (Rizvi et al, 2015; Kumagai et al, 2020a, 2020b). However, only a subset of patients with cancer benefits from ICB therapy, and patients exhibit a variable response to it across cancer types (Carbognin et al, 2015). Therefore, additional studies are needed to understand the influence of the tumor microenvironment (TME) and its constituents on ICB therapy response.

Cancer-associated fibroblasts (CAFs) are a major component of the TME and accumulate in the tumor stroma across multiple cancers (Kalluri, 2016; Kobayashi et al, 2019; Miyai et al, 2020; Piersma et al, 2020). Recent single-cell sequencing analyses have revealed that CAFs can be segregated into several clusters based on their transcriptome (Öhlund et al, 2017; Costa et al, 2018; Lambrechts et al, 2018; Elyada et al, 2019; Kieffer et al, 2020). Major defined subpopulations of CAFs, referred to as myofibroblastic CAFs (myCAFs), inflammatory CAFs (iCAFs), and antigen-presenting CAFs (apCAFs), were first described in pancreatic cancer (Öhlund et al, 2017; Elyada et al, 2019). Single-cell analysis of tumor stroma provided evidence of similar CAF populations in other cancer types, such as breast and lung cancer (Costa et al, 2018; Lambrechts et al, 2018; Kieffer et al, 2020). CAFs are now understood to be a major source of immunosuppressive activity in the TME (Barrett & Puré, 2020; Baker et al, 2021). A pioneering study showed that the CAF-S1 subset, which is characterized by $\alpha$-smooth muscle actin ($\alpha$-SMA) and fibroblast activation protein (FAP) expression, is crucial for the induction of regulatory T cells to promote cancer progression and immunotherapy resistance (Costa et al, 2018; Kieffer et al, 2020). Another study revealed that the infiltration of CAFs expressing leucine-rich repeat-containing 15 (LRRC15), whose expression was induced by TGF-$\beta$, correlated with poor response to ICB therapy across multiple cancer types (Dominguez et al, 2020). Other studies

---

[1]Department of Pathology, Nagoya University Hospital, Nagoya, Japan    [2]Department of Clinical Oncology and Chemotherapy, Nagoya University Hospital, Nagoya, Japan    [3]Department of Immunology, Nagoya University Hospital, Nagoya, Japan    [4]Department of Respiratory Medicine, Nagoya University Hospital, Nagoya, Japan    [5]Center for Advanced Medicine and Clinical Research, Nagoya University Hospital, Nagoya, Japan    [6]Department of Thoracic Surgery, Nagoya University Graduate School of Medicine, Nagoya, Japan    [7]Division of Cancer Immunology, Research Institute/Exploratory Oncology Research and Clinical Trial Center (EPOC), National Cancer Center, Tokyo, Japan

Correspondence: enomoto@iar.nagoya-u.ac.jp
Naoya Asai's present address is Department of Pathology, Fujita Health University Graduate School of Medicine, Toyoake, Japan
Tetsuro Taki's present address is Department of Pathology and Clinical Laboratories, National Cancer Center Hospital East, Kashiwa, Japan
Yoshinori Hasegawa's present address is National Hospital Organization Nagoya Medical Center, Nagoya, Japan
Masahide Takahashi's present address is International Center for Cell and Gene Therapy, Fujita Health University, Toyoake, Japan

---

have consistently indicated that TGF-$\beta$ signaling in CAFs is correlated with immune evasion and immunotherapy failure (Chakravarthy et al, 2018; Mariathasan et al, 2018). However, a complete picture of the roles of diverse CAFs in tumor immunity and responses to ICB is still lacking. It is also unclear whether a specific CAF subset enhances the efficacy of ICB therapy.

We recently described a novel CAF subset characterized by the expression of Meflin (also known as ISLR), a glycosylphosphatidylinositol (GPI)-anchored membrane protein, in pancreatic and colorectal cancers (Mizutani et al, 2019; Kobayashi et al, 2021; Takahashi et al, 2021; Ichihara et al, 2022). Histological and single-cell analyses demonstrated that Meflin-positive (Meflin+) CAFs are weakly positive or negative for $\alpha$-SMA mRNA and are distinct from conventional strongly $\alpha$-SMA-positive CAFs (Mizutani et al, 2019). Analyses of mouse tumor models and human tissue samples suggested that the function of Meflin+ CAFs is the suppression, and not progression, of cancer (Mizutani et al, 2019; Kobayashi et al, 2021). Biochemical analyses showed that Meflin binds to bone morphogenetic protein 7 (BMP7) to augment its signaling, which is known to inhibit the activity of TGF-$\beta$. This suggests that Meflin suppresses various TGF-$\beta$–induced responses, such as tissue fibrosis (Hara et al, 2019; Nakahara et al, 2021). Based on these findings, we propose that Meflin is a specific marker of tumor-restraining CAFs (rCAFs), the existence of which has been postulated previously (Lee et al, 2014; Rhim et al, 2014; Shin et al, 2014; Özdemir et al, 2014). However, the role of Meflin+ CAFs in the tumor response to ICB therapy remains unclear.

In the present study, we showed that the proportion of Meflin+ CAFs in tumor stroma correlated with a favorable response to ICB therapy in patients with non-small cell lung cancer (NSCLC) and mouse syngeneic tumor models. NSCLC tumors with a high number of Meflin+ CAFs exhibited increased CD4+ T-cell infiltration and areas of tumor vessels, suggesting the involvement of Meflin+ CAFs in multiple aspects of the TME. To our knowledge, this is the first study revealing the presence of CAFs that enhance the response to ICB therapy.

# Results

### Meflin is a marker of CAFs present in the stroma of invasive NSCLC tumors

We first examined the expression of Meflin in human lung adenocarcinoma (LUAD) tissues. RNA in situ hybridization (ISH) analysis revealed no apparent Meflin+ cells in the human lung tissue adjacent to the tumors (Figs 1A and S1). In contrast, many Meflin+ cells appeared in the extensive fibroinflammatory stroma within invasive tumors (INV) (Figs 1A and S1). Interestingly, Meflin+ stromal cells were not observed in noninvasive tumors (adenocarcinoma in situ; AIS), whereas they were sparsely present in preinvasive lesions (PIL) with a lepidic growth, a pattern of noninvasive cell proliferation along preexisting alveolar wall, adjacent to invasive tumors (Figs 1A and S1). Meflin expression was also observed in stromal cells that accumulate in tumors developed in an autochthonous LUAD mouse model (KP mice), harboring K-ras$^{G12D}$ and p53 null alleles, after the administration of adenovirus-expressing Cre recombinase (DuPage et al, 2009; Taki et al, 2020), whereas it was hardly detected in the

normal or tumor adjacent tissue (Fig 1B). Statistical analyses showed that the prevalence of Meflin+ cells was positively correlated with an increase in the invasiveness of both human and mouse LUAD tumors (Fig 1C and D). Further fluorescent ISH experiments showed no Meflin expression in E-cadherin+ epithelial cells, including tumor cells, CD31+ endothelial cells, or leukocyte common antigen (LCA)+ leukocytes (Fig S2A). Meflin expression was observed to a varying degree in cells positive for CAF marker proteins, such as collagen type I $\alpha$ 1 (COL1A1), $\alpha$-SMA, and podoplanin (PDPN) (Fig S2B). These data showed that Meflin is a marker of CAFs that accumulate in the invasive stages of both human and mouse LUAD. CAF-specific expression of Meflin was also confirmed by the analysis of single-cell RNA sequencing data of whole cells isolated from human NSCLC and distal non-malignant lung samples (ArrayExpress accession numbers E-MTAB-6149 and E-MTAB-6653, Lambrechts et al, 2018) (Fig 1E).

We observed that CAFs accumulating in human NSCLC exhibited variable expression of Meflin and other CAF markers at the mRNA level (Figs 1F and S2C and D). Duplex ISH staining showed that a substantial fraction of CAFs positive for platelet-derived growth factor receptor $\alpha$ (PDGFR$\alpha$), an established fibroblast marker, was also positive for Meflin. In contrast, the expression of $\alpha$-SMA, a marker of myCAFs (Öhlund et al, 2017; Elyada et al, 2019), was inversely correlated with Meflin expression; ~12% of CAFs expressing $\alpha$-SMA were positive for Meflin, indicating that CAFs with high Meflin expression exhibited low or negative $\alpha$-SMA expression (Figs 1F and S2C and D). Meflin was expressed in ~33% and 12% of FAP+ and PDPN+ CAFs, respectively. Further analysis focusing on the fibroblast cluster of the single-cell RNA sequencing data (Lambrechts et al, 2018) confirmed that Meflin expression was enriched in $ACTA2^{low/neg}$, $IL6^{low/neg}$, or $HLA$-$DRA^{low/neg}$ subsets, indicating that Meflin+ CAFs represent a CAF subset distinct from myCAF, iCAF, and apCAF (Fig 2). These observations suggest that Meflin is a marker of PDGFR$\alpha^{+/-}$ $\alpha$-SMA$^{low/neg}$ FAP$^{+/-}$ PDPN$^{low/neg}$ CAFs in human NSCLC.

### Heterogeneous expression of Meflin in CAFs among patients with NSCLC

Given that previous studies have shown that the expression of Meflin, an rCAF marker, is heterogeneous among patients with pancreatic and colorectal cancers (Mizutani et al, 2019; Kobayashi et al, 2021), we evaluated the prevalence of Meflin+ CAFs in patients with NSCLC who did not receive ICB treatment. To this end, we investigated Meflin expression by ISH in NSCLC samples surgically resected at our institution. As observed in LUAD samples, Meflin expression was explicitly observed in CAFs in the stroma of lung squamous cell carcinoma (LUSC) tissues (Fig 3A). To quantify Meflin+ CAFs, we assigned all stromal cells with oval- to spindle-shaped nuclei as CAFs, excluding lymphocytes, erythrocytes, endothelial cells, and macrophages, based on their morphologies revealed by hematoxylin counterstain. Then, we semiquantitatively scored the expression of Meflin in each patient according to the percentage of Meflin+ CAFs in total CAFs as described in the Materials and Methods section. Interestingly, the analysis revealed that patients with NSCLC showed a two-peak distribution with Meflin scores at the peaks of 5 and 25, respectively (Fig 3B). After the data and the criterion described previously (Mizutani et al, 2019), we set the threshold of Meflin-high as

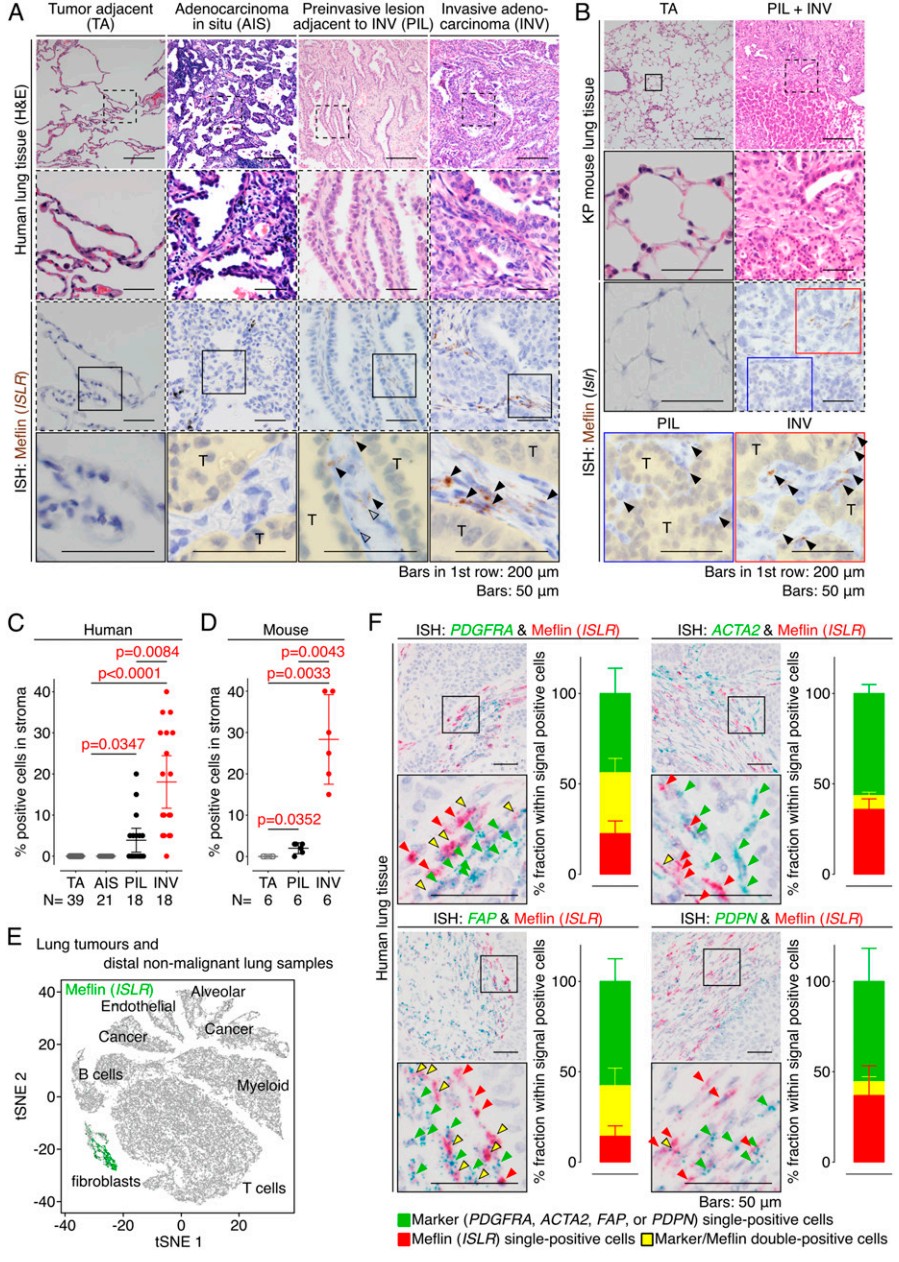

**Figure 1. Meflin is a marker of cancer-associated fibroblasts (CAFs) that appear in invasive human non-small cell lung cancer (NSCLC).**

**(A)** Tissue sections derived from noninvasive (AIS) and invasive (INV) LUAD tumors were stained for H&E (top panels) and Meflin mRNA by ISH (lower panels). Preinvasive lesions adjacent to the invasive tumor (PIL) and tumor adjacent tissue (TA) were also examined. Boxed areas were magnified in lower panels. Black and open arrowheads denote Meflin+ stromal cells and Meflin− macrophages that phagocytize foreign material, respectively. Areas filled with yellow indicate tumor parenchyma comprised of tumor cells (T). **(B)** Preinvasive (PIL) and invasive (INV) lesions of a tumor developed in the autochthonous LUAD mouse model (KP mice) were examined for Meflin expression by ISH. Meflin+ stromal cells (arrowheads) were found in the INV and PIL lesion but not in tumor adjacent tissue (TA). **(C, D)** Quantification of the percentage of Meflin$^+$ cells in all cells with oval- to spindle-shaped nuclei found in the stroma of the tumor adjacent (TA), noninvasive (AIS), preinvasive (PIL), and invasive (INV) cases and area of human LUAD cases (C) and the KP mouse model (D), respectively. All stromal cells found in high-power fields (400x) of the indicated number of cases were evaluated. **(E)** Analysis of a single-cell RNA sequencing dataset showed the specific expression of Meflin in fibroblasts in human NSCLC and distal non-malignant lung samples. **(F)** Duplex ISH for Meflin (red) and other CAF markers (green) showed that Meflin is variably co-expressed with other established CAF markers (*PDGFRA*, *ACTA2*, *FAP*, or *PDPN*) in CAFs of human NSCLC. Green, yellow, and red arrowheads denote cells that are single-positive for the indicated CAF markers, double-positive for the indicated CAF markers and Meflin, and single-positive for Meflin, respectively. All stromal cells that were positive for ISH signals in high-power fields from three independent NSCLC tumors were evaluated and quantified. The graphs show the percent fraction of each subset within ISH signal positive cells.
Source data are available for this figure.

20% or more fibroblasts expressing Meflin, and we used this criterion in the experiment shown below (Fig 3B).

We also confirmed that Meflin expression in CAFs in biopsy samples could be evaluated by ISH (Fig S3). However, we found that some biopsy samples did not contain any stromal components, which made it difficult to examine Meflin expression (Fig S3). We considered these subjects as inappropriate samples and excluded them from the study shown below.

### High infiltration of Meflin$^+$ CAFs correlates with favorable response to ICB in patients with NSCLC

Next, we investigated the involvement of Meflin$^+$ CAFs in tumor response to ICB therapy. We conducted a retrospective observational study of 132 patients with NSCLC who had received ICB monotherapy targeting programmed cell death 1 (PD-1) (nivolumab or pembrolizumab) or programmed cell death 1 ligand 1 (PD-L1) (atezolizumab) at our institution (Fig 4A). The patients were divided into Meflin-high (≥20% Meflin$^+$ CAFs) and Meflin-low (<20% Meflin$^+$ CAFs) groups by ISH analysis (Fig 4B) following the criterion described above. We also evaluated the average total numbers of fibroblasts based on cell morphology, and found that they were comparable between the Meflin-high and Meflin-low groups (Fig 4C). A total of 98 patients were analyzed for outcomes, including objective response rate (ORR) assessed by immunotherapy Response Evaluation Criteria in Solid Tumors (iRECIST) (Seymour et al, 2017), overall survival (OS), and progression-free survival (PFS) (Fig 4A and Table 1). The exclusion criteria are described in Fig 4A.

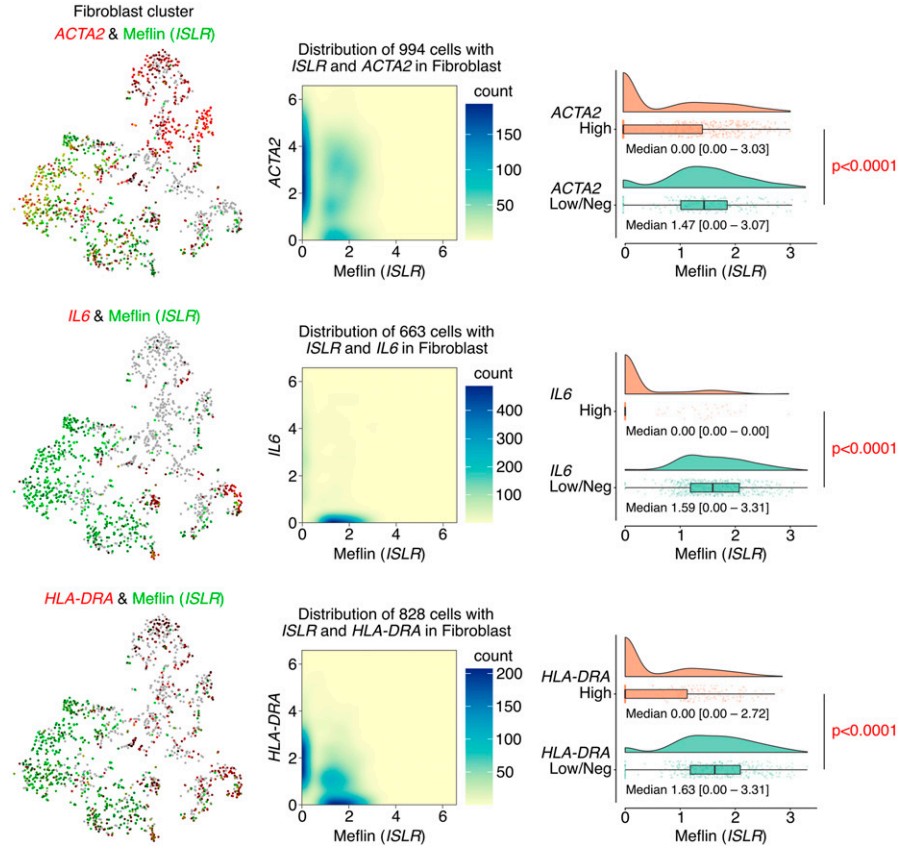

**Figure 2. Meflin⁺ cancer-associated fibroblasts (CAFs) constitute a CAF subset distinct from known CAF subsets.**
t-distributed stochastic neighbor embedding (tSNE) plots obtained by analysis of a single-cell RNA sequencing dataset showed differential expression of Meflin in *ACTA2*⁺, *IL6*⁺, and *HLA-DRA*⁺ CAF subsets, which represent myCAFs, iCAFs, and apCAFs, respectively, of human non-small cell lung cancer and distal non-malignant lung samples (left panels). Density plots (middle panels) and violin/box/scatter plots (right panels) derived from gene expression data of the single-cell analysis showed that Meflin⁺ CAFs constitute a unique CAF subset distinct from myCAF, iCAFs, and apCAFs.
Source data are available for this figure.

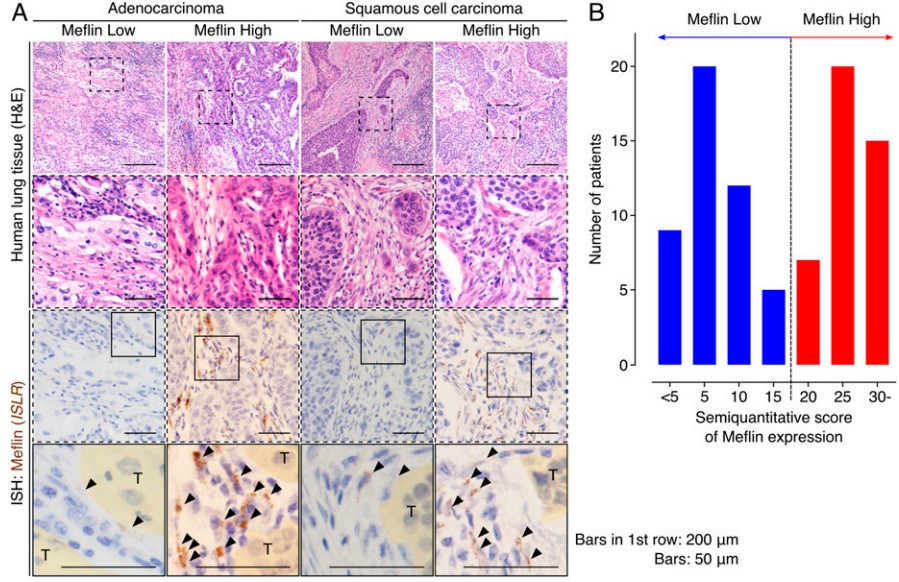

**Figure 3. Meflin expression in cancer-associated fibroblasts (CAFs) is heterogeneous and shows a two-peak distribution across patients with non-small cell lung cancer.**
**(A)** Representative images of Meflin-high and Meflin-low cases of invasive adenocarcinoma (left) and squamous cell carcinoma (right). Dashed boxed areas were magnified in middle panels, which showed serial sections stained for H&E and Meflin mRNA by ISH. In the images of Meflin ISH, boxed areas were magnified in the lowest panels. Areas filled with yellow denote tumor cells (T). Arrowheads indicate Meflin⁺ CAFs. **(B)** Distribution of patients with non-small cell lung cancer stratified by the percentage of Meflin⁺ CAFs in all stromal cells. The number of Meflin⁺ CAFs was counted in randomly selected five high-power microscopic fields. The proportion of Meflin⁺ CAFs was represented as the ratio of Meflin⁺ CAFs to all stromal cells with oval to spindle-shaped nuclei. Meflin-high was defined as ≥20% of stromal cells positive for Meflin.
Source data are available for this figure.

Interestingly, the data showed that ORR of the Meflin-high group (40.3%, 25 of 62 patients) was significantly higher than that of the Meflin-low group (0%, 0 of 32 patients) ($P < 0.0001$, Fig 4D). The threshold of 20% Meflin positivity in all CAFs was found to be the best criterion for predicting response to ICB monotherapy, with an area under the receiver operating characteristic curve (AUC) of 0.632 (95% CI, 0.526–0.738) (Fig 4E). Kaplan–Meier survival analyses using the log-rank Mantel–Cox test revealed that the Meflin-high group had a significantly favorable prognosis in both OS ($P = 0.0281$) and PFS ($P = 0.0011$) than the Meflin-low group (Fig 4F). The analysis

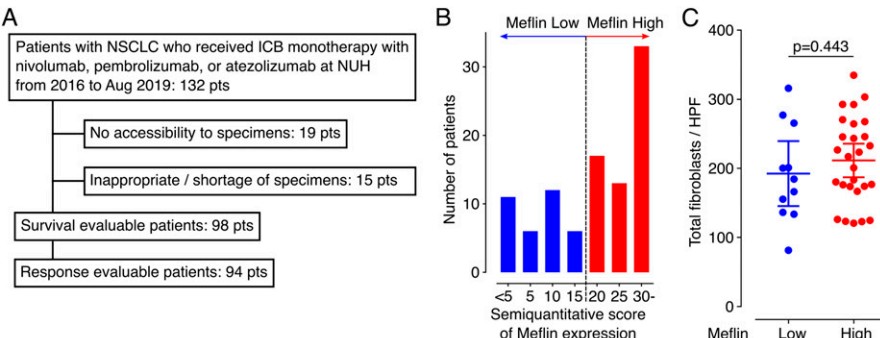

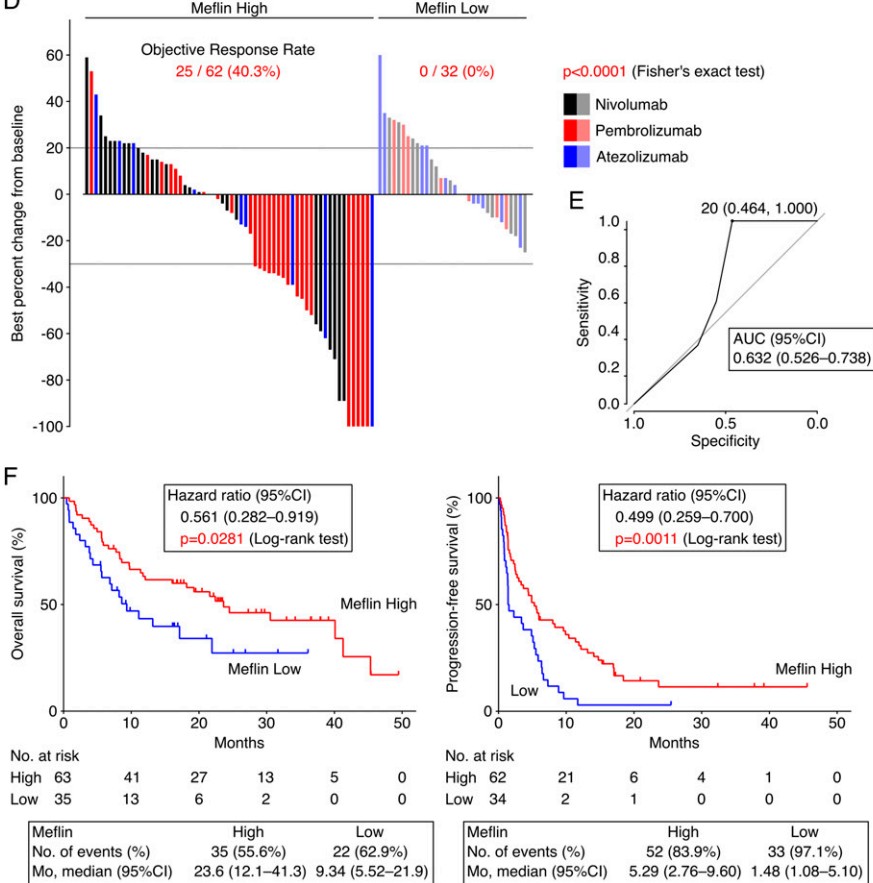

**Figure 4. Non-small cell lung cancer (NSCLC) patients with high Meflin⁺ cancer-associated fibroblast (CAF) infiltration exhibit favorable responses to immune checkpoint blockade (ICB) therapy.**
**(A)** Flow diagram of the selection of eligible patients with NSCLC who received ICB monotherapy for objective response rate analysis in our institution. **(B)** Distribution of patients with NSCLC stratified by the proportion of Meflin⁺ CAFs determined by ISH analysis. The number of Meflin⁺ CAFs was counted in randomly selected four high-power microscopic fields. The proportion of Meflin⁺ CAFs was calculated as the ratio of Meflin⁺ cells to all stromal cells with a spindle morphology. Meflin-high was defined as ≥20% of stromal cells stained for Meflin by ISH. **(C)** Total numbers of fibroblasts found in tissue sections obtained from Meflin-high and -low groups were counted and quantified. NSCLC cases whose surgical specimens were available (n = 38) were evaluated. HPF, high-power field. **(D)** A waterfall plot showing changes in tumor size from baseline determined according to iRECIST criteria in Meflin-high (left) and Meflin-low (right) patients with NSCLC who receive ICB monotherapy. Black, red, and blue bars indicate patients treated with nivolumab, pembrolizumab, and atezolizumab, respectively. **(E)** A receiver operating characteristics (ROC) curve for the percentage of Meflin⁺ CAFs in tumor stroma. The ROC curve was obtained by plotting sensitivity against specificity at each threshold setting. The area under the curve (AUC) (0.632; 95% CI, 0.526–0.738) shown in the plot summarizes the performance of Meflin⁺ CAFs in tumor stroma. **(F)** OS (left) and progression-free survival (right) of Meflin-high (red) and Meflin-low (blue) NSCLC patients treated with ICB therapy. The Meflin-high group showed a favorable response to ICB therapy compared with the Meflin-low group. Shown in the boxes below the plots are the observed numbers of events (deaths or disease progression) and median survival (months) of Meflin-high and Meflin-low groups over the follow-up periods. Mo, months.
Source data are available for this figure.

of the data using a multivariate Cox proportional hazard regression model showed a positive correlation between the percentage of Meflin⁺ CAFs and the outcomes (Table 2). Consistent with recent studies that showed that PD-L1 tumor proportion score (TPS) does not necessarily predict the response to ICB therapy (Carbone et al, 2017; Shen & Zhao, 2018; Lu et al, 2019), the analysis of PD-L1 expression in tumor cells, which was obtained with the 22C3 clone on the Dako Autostainer Link 48 platform, showed that PD-L1 TPS did not correlate with ORR, OS, or PFS in our NSCLC cohort (TPS < 1%; 22.7% ORR, TPS 1–49%; 14.3%, and TPS ≤ 50%; 38.2%) (*P* = 0.071, Fig S4A–C and Table 2). These data demonstrate that Meflin expression in CAFs is a predictive marker for the response to ICB in patients with NSCLC.

## Meflin expression in CAFs correlates with high infiltration of CD4⁺ T cells and tumor vessel area

We next examined the correlation between Meflin expression in CAFs and the profiles of tumor-infiltrating lymphocytes (TILs) using an automated imaging system and a user-trainable image analysis software as described in the Materials and Methods section. Seven-color multiplex immunofluorescence (IF) staining of the specimens of 32 surgically resected NSCLC tumors who received ICB monotherapy revealed that the number of CD4⁺ T cells infiltrating the stroma (interstitium), but not the tumor parenchyma, was significantly higher in Meflin-high patients than in Meflin-low patients (Fig 5A). In contrast, the numbers of CD8⁺ T cells and

**Table 1. Characteristics of patients who received immune checkpoint blockade monotherapies.**

| Variable | Meflin expression | | P-value[a] |
| --- | --- | --- | --- |
| | High | Low | |
| Median age–yr (range) | 70 (42–85) | 69 (43–80) | 0.920[b] |
| Sex–no. (%) | | | 0.663 |
| Female | 23 (36.5%) | 11 (31.4%) | |
| Male | 40 (63.5%) | 24 (68.6%) | |
| Subtype–no. (%) | | | 0.0813 |
| Squamous cell carcinoma (LUSC) | 17 (27.0%) | 3 (8.6%) | |
| Adenocarcinoma (LUAD) | 37 (58.7%) | 27 (77.1%)[c] | |
| Others | 9 (14.3%) | 5 (14.3%) | |
| Targetable EGFR mutation in LUAD–no. (%) | | | >0.999 |
| Not detected | 26 (70.3%) | 17 (68.0%)[c] | |
| Detected | 11 (29.7%) | 8 (32.0%) | |
| PD-L1 TPS[d]–no. (%) | | | 0.139 |
| <1% | 16 (26.2%) | 7 (20.6%) | |
| 1–49% | 18 (29.5%) | 17 (50.0%) | |
| 50%≤ | 27 (44.3%) | 10 (29.4%) | |
| Brinkman index[e]–no. (%) | | | 0.525 |
| <400 | 26 (41.3%) | 12 (34.3%) | |
| 400≤ | 37 (58.7%) | 23 (65.7%) | |
| ECOG-PS–no. (%) | | | 0.255 |
| 2≤ (Poor) | 8 (12.7%) | 8 (22.9%) | |
| 0 or 1 (Good) | 55 (87.3%) | 27 (77.1%) | |
| Tx line[f]–no. (%) | | | **0.0046** |
| 1L or 2L | 46 (73.0%) | 15 (42.9%) | |
| 3L≤ | 17 (27.0%) | 20 (57.1%) | |

[a]Fisher's exact test.
[b]Mann–Whitney U test.
[c]One patient with anaplastic lymphoma kinase (ALK)-rearranged non-small cell lung cancer.
[d]Tumor Proportion Score was the percentage of viable tumor cells showing partial or complete membrane staining at any intensity.
[e]Brinkman index was defined as the number of cigarettes smoked per day multiplied by the number of years of smoking (e.g., if one smoked 20 cigarettes per day for 20 yr, the Brinkman index is 400).
[f]Only conventional cytotoxic chemotherapeutics were considered in the treatment line.
P-values in bold showed statistically significant differences.

CD4[+]FoxP3[+] regulatory T cells were comparable between the two groups (Fig 5A). There was also no difference in the numbers of CD45RO[+] memory CD4[+] T cells, CD45RO[+] memory CD8[+] T cells, or CD20[+] B cells in both the stroma and tumor parenchyma between the two groups (Fig S5).

Our previous study showed that tumors developed in the pancreas of Meflin KO mice exhibited a decrease in tumor vessel area accompanied by changes in collagen configuration (Mizutani et al, 2019). Higher tumor vascularity is also associated with better tumor responses to ICB therapy in mouse models (Zheng et al, 2018). Immunostaining of the NSCLC tumor samples with anti-CD31 antibody showed that the Meflin-high group tumors had greater tumor vessel area than the Meflin-low group (Fig 5B). These data suggest that the infiltration of Meflin[+] CAFs is associated with increased tumor vessel perfusion.

### Defective response of tumors to ICB therapy in Meflin-KO mice

Next, we determined whether Meflin expression in CAFs is crucial for the response of tumors to ICB using C57BL/6J wild-type (WT) mice and Meflin-KO mice. We previously reported that Meflin-KO mice displayed decreased spleen weight compared with WT mice (Maeda et al, 2016). Therefore, we first examined the immunophenotype of lymphocytes isolated from the spleen of Meflin-KO mice to compare it with that of WT mice. The data showed no differences in the proportions of CD4[+], CD8[+], and regulatory T cells (Fig S6A and B).

Our first attempts on mice bearing tumors of murine Lewis lung carcinoma (LLC) cells were unsuccessful, showing that LLC syngeneic lung tumors developed in C57BL/6J mice were not responsive to either anti-mouse PD-1 (mPD-1) or anti-mouse PD-L1 antibody. We therefore next subcutaneously transplanted syngeneic MC-38 colorectal cancer

**Table 2. Hazard ratios and *P*-values for multivariate Cox proportional hazard regression model analysis in patients with non-small cell lung cancer subjected to immune checkpoint blockade monotherapies.**

| Variable | Hazard ratio (95% CI) for OS | *P*-value for OS | Hazard ratio (95% CI) for progression-free survival | *P*-value for progression-free survival |
|---|---|---|---|---|
| Age | 0.961 (0.934–0.990) | **0.0085** | 0.955 (0.928–0.983) | **0.0018** |
| Subtype | | 0.866[a] | | 0.849[a] |
| LUSC | Reference | | Reference | |
| LUAD | 0.790 (0.334–1.87) | 0.592 | 0.771 (0.289–2.06) | 0.604 |
| Others | 0.847 (0.293–2.45) | 0.760 | 0.884 (0.307–2.55) | 0.819 |
| PD-L1 TPS | | 0.128[a] | | 0.919[a] |
| <1% | Reference | | Reference | |
| 1–49% | 0.979 (0.441–2.17) | 0.958 | 1.06 (0.548–2.04) | 0.868 |
| 50%≤ | 1.84 (0.858–3.94) | 0.117 | 0.927 (0.473–1.82) | 0.825 |
| Targetable EGFR mutation | | | | |
| Not detected | Not included[b] | | Reference | |
| Detected | | | 2.49 (1.06–5.87) | **0.0374** |
| Brinkman index | | | | |
| <400 | Reference | | Reference | |
| 400≤ | 0.398 (0.189–0.836) | **0.0151** | 0.706 (0.340–1.47) | 0.349 |
| ECOG-PS | | | | |
| 2≤ (Poor) | Reference | | Reference | |
| 0 or 1 (Good) | 0.0468 (0.0204–0.108) | **<0.0001** | 0.0994 (0.0437–0.226) | **<0.0001** |
| Tx line | | | | |
| 1L or 2L | Reference | | Reference | |
| 3L≤ | 1.12 (0.604–2.09) | 0.712 | 0.803 (0.450–1.43) | 0.458 |
| Meflin | | | | |
| Low | Reference | | Reference | |
| High | 0.473 (0.242–0.922) | **0.0280** | 0.486 (0.268–0.881) | **0.0174** |

[a]Wald test.
[b]Targetable EGFR mutation status was not included in the multivariate analysis for OS because of the *P*-value in the univariate analysis (*P* = 0.291).
*P*-values in bold showed statistically significant differences.

(CRC) cells, a well-established cell line that is used to study the antitumor effect of ICB therapy (House et al, 2020; Mager et al, 2020), into WT mice and Meflin-KO mice, followed by intraperitoneal administration of anti–mPD-1 antibody or isotype control IgG on day 4, 7, and 10 after transplantation (Fig 6A). WT mice treated with anti–mPD-1 antibody, but not isotype control IgG, had a statistically better prognosis than Meflin-KO mice (Fig 6B). The effect of Meflin deficiency on the antitumor effect of mPD-1 antibody was also evaluated using a linear mixed-effect model with restricted maximum likelihood estimates, which showed that the suppressive effect of anti–mPD-1 antibody on tumor growth was significantly weakened by Meflin-KO (*P* = 0.0041), although Meflin-KO itself did not exhibit altered tumor growth (*P* = 0.901, Fig 6C).

The role of Meflin in promoting the antitumor effect of anti–mPD-1 antibody was also confirmed in another experimental setup, in which we orthotopically transplanted syngeneic EO771 breast cancer (BC) cells (House et al, 2020; Zheng et al, 2018) into the fourth right mammary fat pad of WT mice and Meflin-KO mice, followed by intraperitoneal administration of anti–mPD-1 antibody or control IgG on day 6, 9, and 12 after the transplantation (Fig 6D). On day 19, a suppressive effect of anti–mPD-1 antibody on tumor volumes was observed, which was abrogated in Meflin-KO mice as indicated by the *P*-value of 0.0227 obtained according to the two-sided permutation Brunner–Munzel test with Holm–Bonferroni correction (Fig 6E). The importance of Meflin expression in CAFs upon anti–mPD-1 antibody treatment was measured by the effect size (Cliff's delta) of –0.796 (95% CI –1.00 to –0.184) (Fig 6E). Consistent with the analysis of human NSCLC samples, the tumor vessel area in EO771 tumors developed in WT mice was greater than that in Meflin-KO mice (Fig 6F). Taken together, Meflin expression in CAFs might facilitate the antitumor effect of anti–mPD-1 antibody by increasing the tumor vascular bed.

### Meflin expression in CAFs associates with TIL activation in mice

We then explored the status of TILs in MC-38 tumors developed in WT and Meflin-KO mice (Fig 7A). Unfortunately, the infiltration of

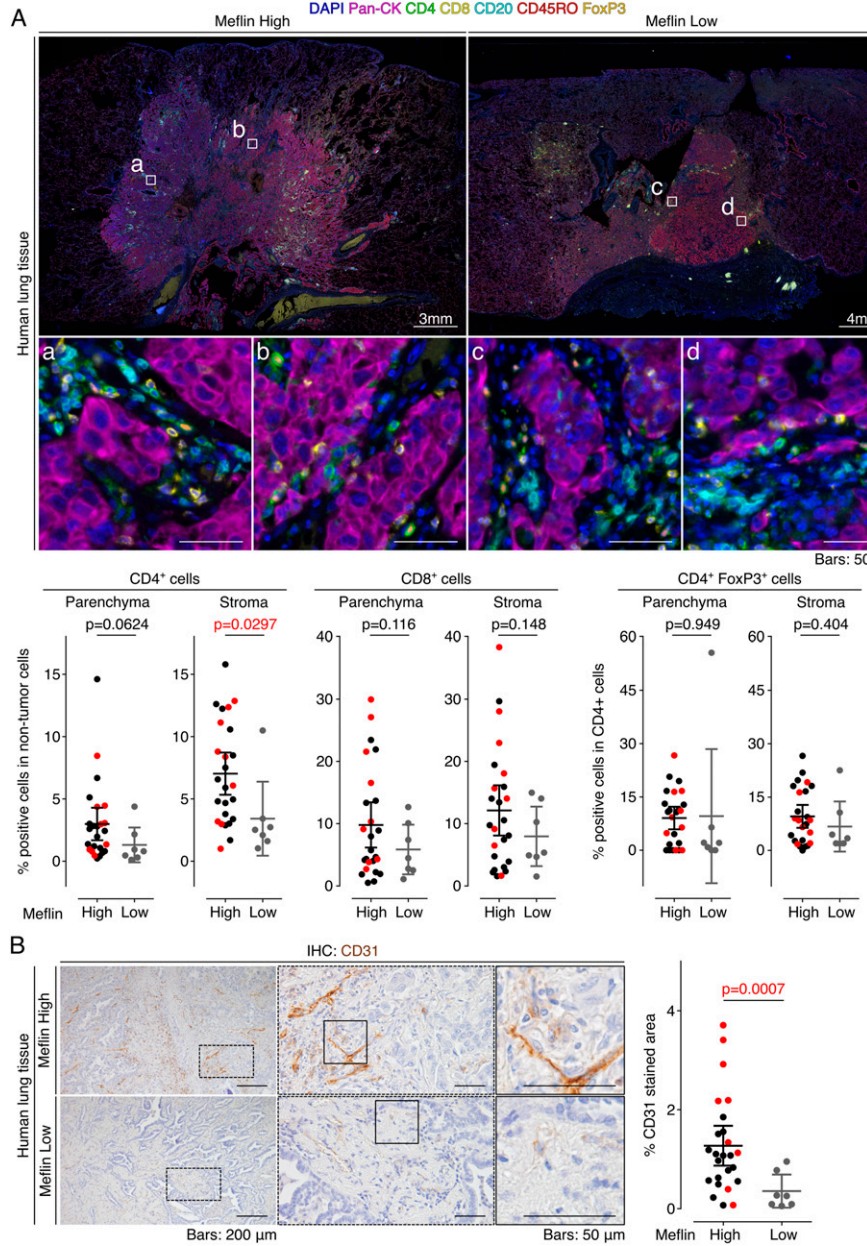

**Figure 5. Meflin expression in cancer-associated fibroblasts correlates with tumor infiltration of CD4⁺ T cells and vascular area in non-small cell lung cancer patients treated with immune checkpoint blockade (ICB) monotherapy.**

**(A)** Tissue sections prepared from tumors of patients with non-small cell lung cancer who received ICB monotherapy were stained by multiplex immunofluorescence for the indicated TIL markers, nuclei, and pan-cytokeratin (Pan-CK), followed by imaging with a multispectral imaging system. Representative images of Meflin-high (left, n = 25) and Meflin-low (right, n = 7) cases are shown. Boxed regions were magnified in lower panels. The lower panels' graphs show the quantification of the percentage of CD4⁺, CD8⁺, and CD4⁺FoxP3⁺ T cells in all TILs infiltrated in intra-tumor (parenchyma) and stroma regions of each group. Red dots in the graphs denote the responders to ICB monotherapy. **(B)** Representative images of tissue sections of Meflin-high (upper panel) and Meflin-low (lower panel) tumors stained for CD31. The graph on the right shows the quantification of tumor vessel areas in each group. High magnification views randomly selected from 25 Meflin-high and 7 Meflin-low cases were analyzed and quantified. Red dots in the graphs denote the responders to ICB monotherapy.

Source data are available for this figure.

CD4⁺ T cells, CD8⁺ T cells, and CD25⁺FoxP3⁺ regulatory T cells varied across two independent experiments. Therefore, we concluded that, contrary to the analysis of human NSCLC tissues, T-cell infiltration was similar between tumors developed in WT and Meflin-KO mice (Fig 7B). Interestingly, we found that the expression of immune checkpoint molecules in some subsets of TILs was higher in tumors of WT mice than that of Meflin-KO mice, which included T-cell immunoglobulin and mucin-domain-containing molecule 3 (TIM-3) on CD8⁺ and regulatory T cells, PD-1, CD25, cytotoxic T-lymphocyte associated protein 4 (CTLA-4), and inducible T-cell co-stimulator (ICOS) on CD8⁺ T cells (Fig 7C–E). Previous studies have indicated that several molecules such as PD-1 and ICOS on CD8⁺ T cells are associated with the activation of antitumor immunity and favorable clinical responses to ICB therapy (Gros et al,

2014, 2016; Xiao et al, 2020; Kumagai et al, 2020b). These exploratory analyses suggest that Meflin expression in CAFs is associated with TIL activation in mice, but not their recruitment or infiltration into tumors.

**Induced expression of Meflin in the lineage of Meflin⁺ cells enhance the antitumor activity of anti–mPD-1 antibody therapy**

The findings described above suggested that the induction of Meflin expression in CAFs can be a therapeutic strategy to enhance ICB therapy efficacy. To prove this, we generated a transgenic mouse line expressing mouse Meflin under the tetracycline response element (TRE) promoter (TRE-Meflin) (Fig 8A). This line was then crossed with Meflin-Cre (Hara et al, 2019, 2021; Mizutani et al,

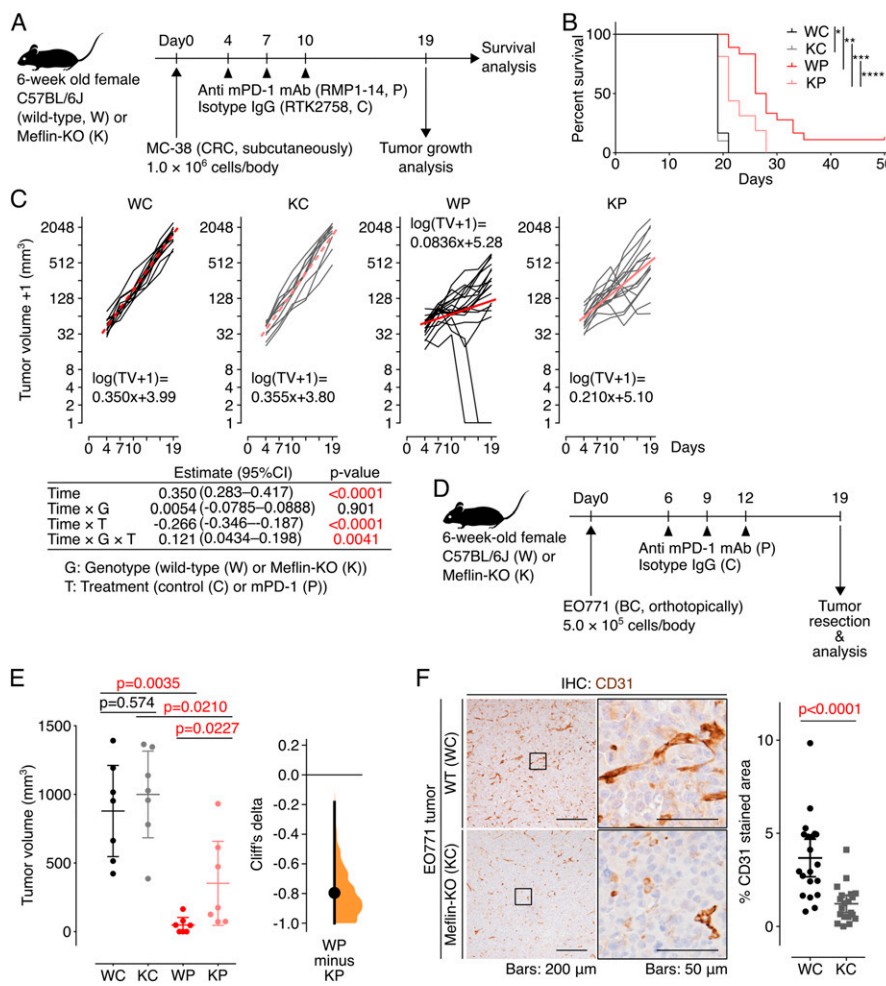

**Figure 6. Tumors developed in Meflin-KO mice exhibit poor response to anti–mPD-1 therapy.**
**(A)** An experimental protocol to test the antitumor effect of anti–mPD-1 therapy in mice. MC-38 mouse CRC cells were transplanted into C57BL/6 wild-type (W) or Meflin-KO (K) mice at day 0, followed by the intraperitoneal administration of anti–mPD-1 (P) or an isotype control antibody (C). **(B)** Survival of wild-type (W) and Meflin-KO (K) mice treated with isotype control (WC and KC, respectively) or treated with anti-mPD-1 antibody (WP and KP, respectively) in mice. The numbers of mice tested for WC, KC, WP, and KP groups were 12, 10, 18, and 16, respectively. *$P$ = 0.658; **$P$ < 0.0001; ***$P$ = 0.0033; ****$P$ = 0.0033. **(C)** Time courses of the volumes of tumors developed in the indicated groups (black and grey lines). For the log transformation of tumor volumes, one was added to every tumor volume. Red lines indicate linear approximations. The table shown under the graphs shows the restricted maximum likelihood estimates of each parameter in a linear mixed-effects model that includes the interactions of time, time and mouse genetic background (G), time and anti–mPD-1 therapy (T), and time and G and T while adding variable effects to the slope and intercept for each individual. **(D, E)** EO771 mouse BC cells were transplanted into wild-type (W) or Meflin-KO (K) mice at day 0, followed by the intraperitoneal administration of anti–mPD-1 antibody (P) or an isotype control antibody (C). The number of mice tested for each group was seven. Shown in (E) are the quantification of tumor volume of each group (left) and the nonparametric estimate of effect size calculated by Cliff's delta (right). **(F)** Tissue sections from EO771 tumors developed in wild-type (WC) and Meflin-KO (KC) were stained for CD31 to visualize tumor vessels (left), followed by the quantification of the stained areas (right). Boxed regions were magnified in adjacent panels.
Source data are available for this figure.

2019) and Rosa26-LSL (LoxP-stop-LoxP)-rtTA3 (third-generation reverse tetracycline-regulated transactivator) mice (JAX stock #029617, Dow et al, 2014) to generate mice with doxycycline-induced expression of Meflin in Meflin-lineage cells. To confirm induced Meflin expression in mice carrying all three alleles (hereafter referred to as Meflin-TO), we administered doxycycline in drinking water (2 mg/ml) to Meflin-TO mice and transplanted MC-38 and EO771 cells subcutaneously and orthotopically, respectively, followed by the analysis of Meflin expression (Fig S7A). Quantitative PCR (qPCR) of the tumor tissue samples revealed that doxycycline induced Meflin expression in *Col1a1*[+] stromal cells, representing CAFs, in tumors developed in Meflin-TO mice administered doxycycline, but not in control mice that lack the Meflin-Cre allele (Fig S7B). ISH and qPCR also confirmed the induced Meflin expression in cultured CAFs isolated from tumors developed in Meflin-TO but not that of control mice (Fig S7C).

Consistent with the analysis of tumor vessel area of human NSCLC tissues and tumors developed in WT and Meflin-KO mice (Figs 5B and 6F), the area of vasculature in tumors developed in doxycycline administered-Meflin-TO was significantly larger than that in Meflin-TO mice not administered doxycycline and control mice that lacked the Meflin-Cre allele (Fig S7D).

Finally, littermates obtained by crossing Meflin-Cre, TRE-Meflin, and Rosa26-LSL-rtTA3 mice were administered doxycycline and

subcutaneously transplanted with MC-38 cells, followed by intra-peritoneal administration of anti–mPD-1 antibody on day 4, 7, and 10. Genotyping of the mice was performed on day 19 (Fig 8B). The genotypes of the mice were blinded to the investigators during data acquisition and analysis. The results showed that Meflin-TO administered doxycycline and anti–mPD-1 antibody exhibited a more favorable prognosis and response than control mice (Fig 8C and D). These data supported the notion that Meflin is a CAF marker and functionally contributes to a subset of CAFs that facilitate the antitumor effect of ICB therapy (Fig 8E).

# Discussion

In the present study, we focused on the role of Meflin, a recently identified rCAF marker in pancreatic and colorectal cancers (Mizutani et al, 2019; Kobayashi et al, 2021), in tumor response to ICB therapy through the analyses of human NSCLC samples and syngeneic tumor mouse models. Our data suggest that Meflin expression in CAFs correlates with favorable tumor response to ICB therapy, leading to the hypothesis that Meflin[+] CAFs promote the host antitumor immune response. Previous studies have shown that immunosuppressive CAFs, such as $\alpha$-SMA[+]FAP[+] CAFs, LRRC15[+] CAFs,

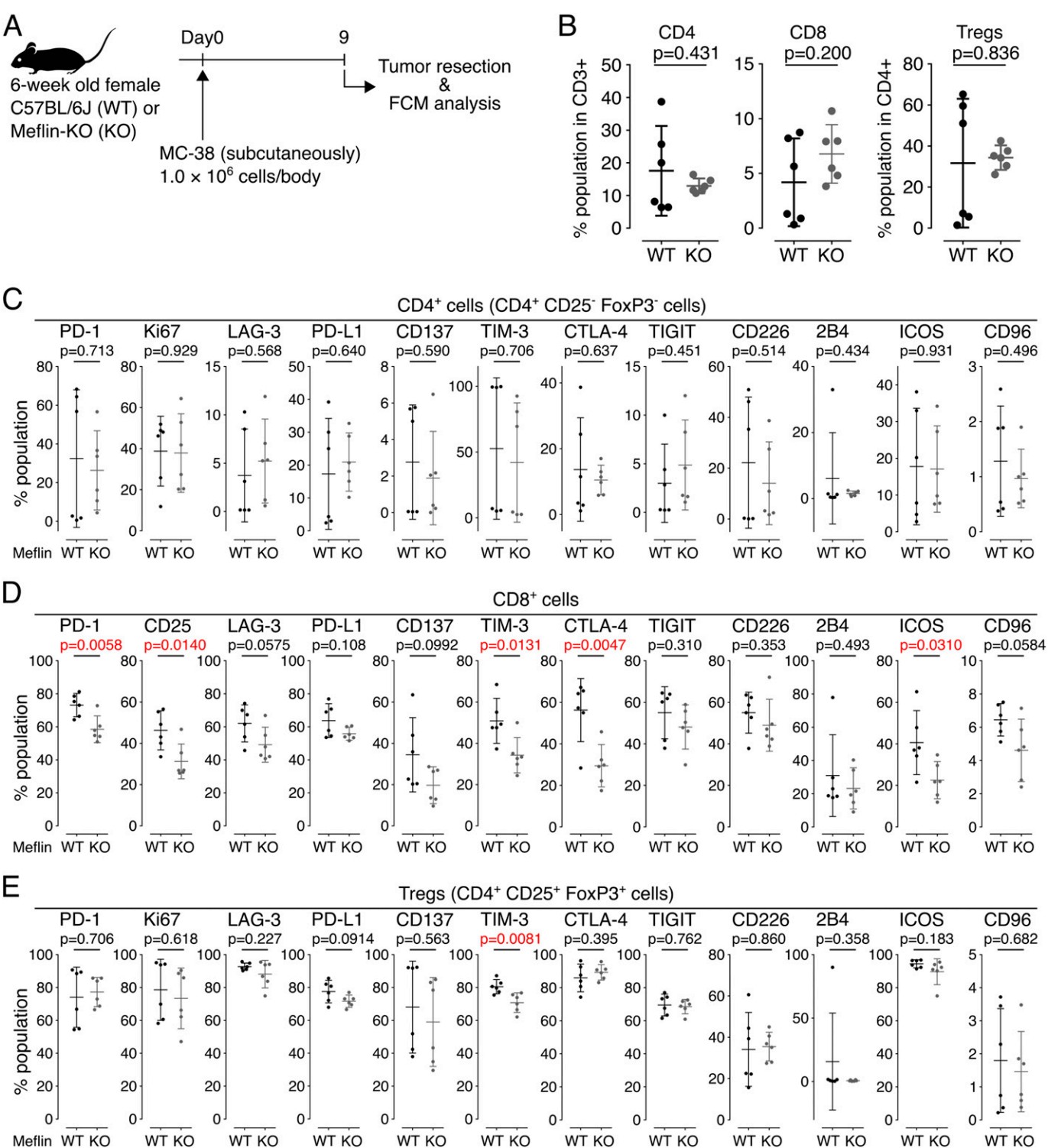

**Figure 7. Exploratory analysis of the expression of various markers on TILs isolated from MC-38 tumors developed in wild-type or Meflin-KO mice.**
**(A, B)** MC-38 cells were transplanted into wild-type (WT) or Meflin-KO (KO) mice on day 0, followed by tumor resection and flow cytometric (FCM) analysis on day 9 (A). For FCM analysis, CD3⁺ T cells were first gated among TILs isolated from the tumors by positive selection, and then these cells were analyzed for expression of the indicated T-cell markers. The graphs on the right show the quantification of CD4⁺ and CD8⁺ T cells and CD4⁺CD25⁺FoxP3⁺ regulatory T cells (Treg) in TILs isolated from tumors developed in WT and KO mice. **(C, D, E)** CD3⁺CD4⁺CD25⁻FoxP3⁻, CD3⁺CD8⁺, and CD4⁺CD25⁺FoxP3⁺ T cells sorted from TILs isolated from tumors developed in wild-type (WT) or Meflin-KO (KO) mice were analyzed for the indicated markers by FCM analysis.
Source data are available for this figure.

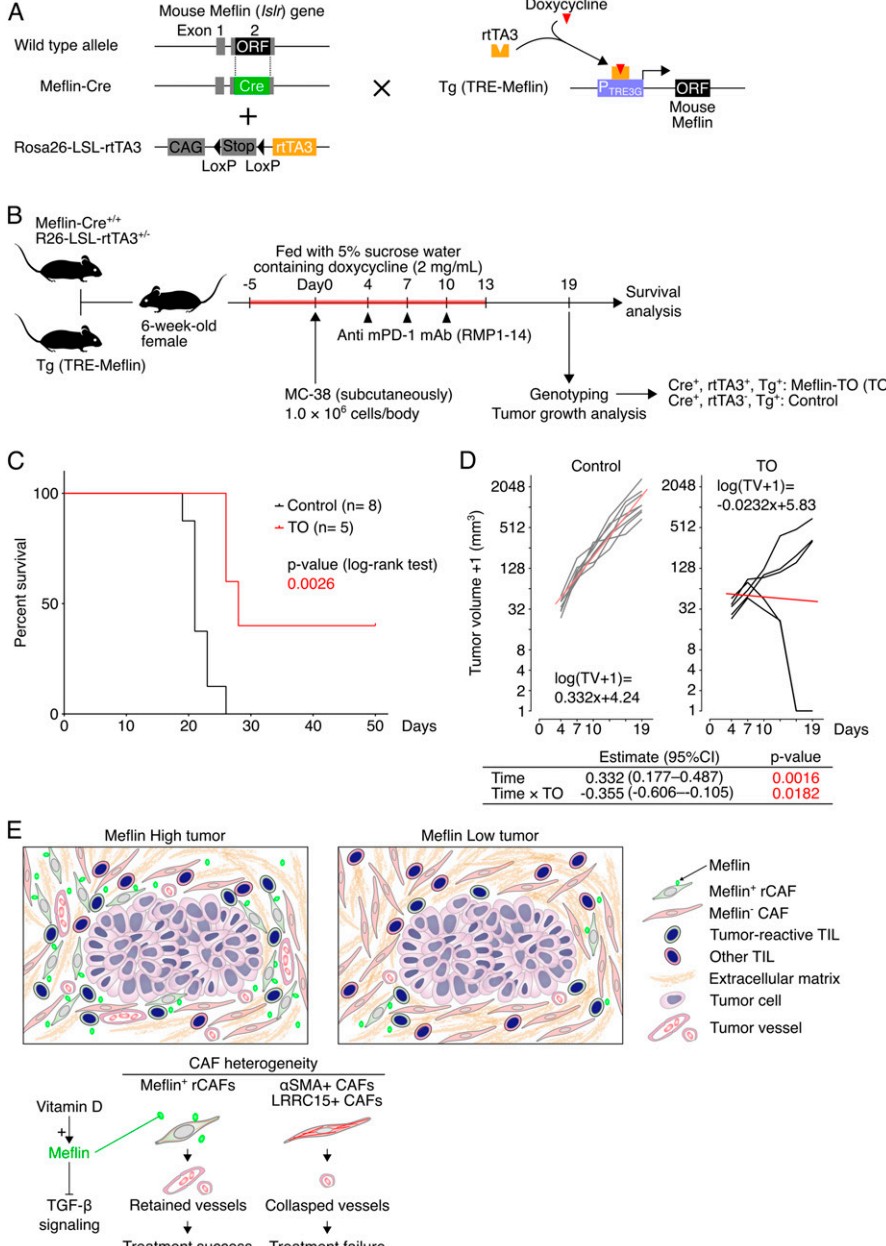

**Figure 8. Induced expression of Meflin in cancer-associated fibroblasts (CAFs) improves tumor response to anti–mPD-1 therapy.**
**(A)** A diagram of the generation of the mouse line carrying Meflin-Cre, which constitutively expresses Cre under the control of the Meflin promoter, Rosa26-LSL-rtTA3, and Tg (TRE-Meflin) alleles that exhibit induced Meflin expression in Meflin-lineage cells upon doxycycline administration. ORF, open reading frame; CAG, chicken β-actin promoter; Stop, stop element; rtTA3, the third-generation reverse tetracycline-regulated transactivator; Tg, transgenic; TRE, tetracycline-response element. **(B)** Meflin-Cre[+/+]; Rosa26-LSL-rtTA3[+/−] mice were crossed with TRE-Meflin transgenic mice. The resultant 6-wk-old female mice fed with doxycycline were subcutaneously implanted with MC-38 cells, followed by anti–mPD-1 therapy, genotyping, and tumor analysis. Mice that harbor all of the Meflin-Cre, Rosa26-LSL-rtTA3, and TRE-Meflin alleles were termed Meflin TO (Tet-on) mice. **(C)** Survival of control (Meflin-Cre[+/−]; Rosa26-LSL-rtTA3[−/−]; TRE-Meflin) and Meflin TO (Meflin-Cre[+/−]; Rosa26-LSL-rtTA3[+/−]; TRE-Meflin) mice treated with anti–mPD-1 antibody. **(D)** Time courses of the volumes of tumors developed in the indicated groups (black and grey lines). For the log transformation of tumor volumes, one was added to every tumor volume. Red lines indicate linear approximations. The table shown under the graphs shows the restricted maximum likelihood estimates of each parameter in a linear mixed-effects model that includes the interactions of time and time and induced Meflin expression (TO) while adding variable effects to the slope and intercept for each individual. **(E)** Graphical summary and working hypothesis for CAF heterogeneity and its role in immune checkpoint blockade response. Our data demonstrated that infiltration of Meflin[+] CAFs was associated with increased tumor vessel area, which may allow blood and immune cells and macromolecules like antibodies to access the tumor. Source data are available for this figure.

and TGF-β-activated CAFs, suppress antitumor immunity and are associated with ICB therapy failure (Chakravarthy et al, 2018; Mariathasan et al, 2018; Dominguez et al, 2020; Kieffer et al, 2020). We propose that a balance between the immunosuppressive CAFs and Meflin[+] rCAFs is crucial for determining the net response to ICB therapy (Fig 8E).

Given our initial data that Meflin expression in CAFs correlated with the favorable outcomes of NSCLC patients treated with ICB, it was an unexpected finding that the repertoire of TILs was almost comparable between Meflin-high and Meflin-low groups, except for CD4[+] T cells. Meflin expression did not affect the infiltration of CD8[+] T cells or regulatory T cells, suggesting that the mechanism of action of Meflin protein and Meflin[+] CAFs may be different from

those occurring with other molecules and modalities that directly boost tumor immunity by regulating the interactions of tumor cells with the host tissue. Interestingly, CD4[+] T-cell infiltration in the stroma, but not in the tumor parenchyma, correlated with the number of Meflin[+] CAFs in patients with NSCLC. These data may provide a mechanistic clue to the cancer-restraining role of Meflin[+] rCAFs. Several studies have shown the involvement of CD4[+] helper T cells in tumor immunity and response to ICB (Zuazo et al, 2019; Kagamu et al, 2020; Li et al, 2020; Liu et al, 2020), which is distinct from their prominent role in inducing tumor-reactive cytotoxic T cells (Kreiter et al, 2015; Sahin et al, 2017; Borst et al, 2018; Zuazo et al, 2020). Notably, the repertoire of TILs was comparable between tumors developed in WT and Meflin-KO mice in a syngeneic tumor

model, suggesting that CD4[+] T-cell infiltration depends on the overall function of Meflin[+] CAFs, but not specifically the function of Meflin.

We previously identified BMP7 as a ligand of Meflin and reported that it augments BMP7 signaling, which suppresses TGF-$\beta$ signaling and tissue fibrosis (Hara et al, 2019; Kobayashi et al, 2021). The present data showed that the tumor vessel areas correlated with Meflin expression in CAFs in both human NSCLC tissues and mouse models. Although not proven in the present study, an intriguing hypothesis is that Meflin-mediated suppression of tissue fibrosis or decrease in interstitial pressure facilitates tumor vessel perfusion and therapeutic antibody delivery. In addition, the involvement of Meflin in controlling the enhanced permeability and retention (EPR) effect, which refers to the ability of macromolecules such as anti–PD-1 antibodies to accumulate in the tumor tissue (Matsumura & Maeda, 1986; Matsumura, 2020), will be a subject of future research.

An appealing feature of Meflin is that none (0%) of the patients with NSCLC in our institution with low Meflin expression in CAFs responded to ICB therapy. These data suggest that the number of Meflin[+] CAFs could be a marker for identifying patients who will not benefit from ICB therapy. The present study also showed that the induced expression of Meflin in CAFs increased the antitumor effect of anti–mPD-1 antibody in a transgenic mouse model. The data implied that the augmentation of Meflin expression in CAFs could be a therapeutic strategy to increase the efficacy of ICB therapy. In relation to this issue, we previously showed that calcipotriol, a vitamin D analog, induced the up-regulation of Meflin expression in CAFs isolated from human pancreatic cancer (Mizutani et al, 2019). Our findings are consistent with a previous study that showed that calcipotriol administration induced changes in the TME with an increase in the delivery of chemotherapeutic agents to the tumor in a pancreatic cancer mouse model (Sherman et al, 2014). Ongoing clinical trials have investigated the use of vitamin D analogs in combination with immune checkpoint inhibitors and chemotherapeutic agents (Gong et al, 2018). It would be interesting to study how Meflin is involved in vitamin D–mediated remodeling of the TME and the increased efficiency of ICB therapy in clinical settings.

In conclusion, we identified a CAF subset marked by Meflin expression and found its prevalence to be associated with a favorable response to ICB therapy in patients with NSCLC and syngeneic tumor mouse models. Induction of Meflin expression in CAFs augmented the tumor response to ICB therapy in mice. Together with other studies that identified CAF subsets that suppress antitumor immunity and are associated with ICB treatment failure, we propose that the heterogeneity of CAFs determines the net response of tumors to ICB therapy.

# Materials and Methods

### Subject details

This study was conducted in accordance with the Declaration of Helsinki and approved by the Ethics Committee of Nagoya University Graduate School of Medicine (approval number 2017–0127-3). We retrospectively enrolled a cohort of patients with NSCLC at Nagoya University Hospital to identify the effects of Meflin[+] CAFs on patient response to ICB monotherapy. The cohort included patients with

advanced or recurrent NSCLC who received programmed cell death 1 (PD-1) or programmed cell death 1 ligand 1 (PD-L1) antibody-based ICB monotherapy (n = 132, Table 1). All patients consented to the Institutional Review Board-approved protocols permitting specimen collection.

### Human tumor samples

98 of 132 patients from the cohort were included for further investigation because of their characteristics described in Fig 4A. 38 of 98 selected patients from the cohort were obtained at the time of surgery, and 60 tumor tissues from the cohort were obtained at the time of diagnostic biopsy or re-biopsy before ICB monotherapy.

### Visualization of previously processed single-cell RNA seq dataset

To visualize Meflin expression in the single-cell RNA sequencing dataset (ArrayExpress accession numbers E-MTAB-6149 and E-MTAB-6653, Lambrechts et al, 2018), we used the web-based visualization tool SCope (https://gbiomed.kuleuven.be/scRNAseq-NSCLC, Davie et al, 2018). We also analyzed the dataset via BBrowser (BioTuring, Le et al, 2020 *Preprint*) to extract data of gene expression in fibroblasts and divide them into *ACTA2*, *IL6*, or *HLA-DRA*-high (normalized value: 1–) or -low/neg (normalized value: <1) groups, followed by processing the data with the R package (v.4.1.2, https://www.r-project.org/) to obtain density and violin plots using the tidyverse function implemented in the R package (Wickham et al, 2019). Given that the data showed non-normal distribution and heteroscedasticity, we chose a nonparametric Brunner–Munzel test to examine the expression data in each group.

### Clinical efficacy analysis

OS was defined as the time from the start of ICB therapy until death from any cause. PFS was defined as the time from ICB monotherapy initiation until disease progression or death from any cause. Patient follow-up ended when an outcome was recorded or censored as of the database lock on 30 April 2020. Response to ICB was determined according to immunotherapy Response Evaluation Criteria in Solid Tumors (iRECIST) at each time point, which included iCR (complete response), iSD (stable disease), and iPR (partial response), as well as unconfirmed PD (iUPD) and confirmed PD (iCPD). ORR was defined as the ratio of patients who achieved iPR or iCR.

### Animals

All mice were kept in specific pathogen-free conditions in the Division of Experimental Animals, Nagoya University Graduate School of Medicine. All experimental protocols were approved by the Animal Care and Use Committee of Nagoya University Graduate School of Medicine. The generation of an autochthonous lung adenocarcinoma mouse model (KP mice), Meflin-KO mice, and Meflin-Cre mice have been described previously (Maeda et al, 2016; Mizutani et al, 2019; Taki et al, 2020; Hara et al, 2021).

We generated a transgenic mouse line carrying a third-generation tetracycline-response element (TRE)-Meflin to induce Meflin expression in CAFs. To this end, the open reading frame of

the mouse Meflin gene was inserted into the multiple cloning site of a tetracycline-inducible expression vector pTRE3G (631168; Clontech), followed by microinjection into fertilized eggs of C57BL/6 mice (RRID: IMSR_JAX:000664). We established four lines of TRE-Meflin with germline transmission. We selected one line for further experiments based on the confirmed doxycycline-mediated induction of Meflin expression in mouse embryonic fibroblasts. The TRE-Meflin mice were crossed with Meflin-Cre and Rosa26-CAGs-LSL (LoxP-Stop-LoxP)-rtTA3 knock-in mice (hereafter termed R26-rtTA3, JAX stock# 029617, RRID: IMSR_JAX:029617).

Genomic DNA extracted from mouse tails was used for PCR-based genotyping. The primer sequences were as follows: Meflin-KO forward, 5′-GCTGCATTTGAGCTGAGCCTCTGG-3′; Meflin-KO reverse, 5′-AACCCCTTCCTCCTACATAGTTGG-3′; Meflin-Cre forward, 5′-TAGGTGG-TATTGGATTCTGGCTGGG-3′; Meflin-Cre reverse, 5′-TTGAAGTAGTCGAC-GATGTCCTGG-3′; R26-rtTA3 forward, 5′-TACTCAATGGAGTCGGTA TCGAAGGC-3′; R26-rtTA3 reverse, 5′-CCAATACGCAGCCCAGTGTAAA GTGG-3′; TRE-Meflin forward, 5′-GATCGCCTGGAGCAATTCCACAAC-3′; TRE-Meflin reverse, 5′-CTGTTGGCTGACAGGCTCAGTGTGG-3′. The PCR product sizes from Meflin-KO, Meflin-Cre, R26-rtTA3, and TRE-Meflin alleles were 267, 385, 393, and 315 bp, respectively.

## Cell lines

MC-38 (ENH204; Kerafast), a murine colon cancer cell line, and EO771 (94A001; CH3 BioSystems, RRID: CVCL_GR23), a murine breast cancer cell line, were maintained in DMEM (Gibco) and RPMI 1640 (Gibco), respectively, supplemented with 10% heat-inactivated FBS. All cell lines were routinely screened for mycoplasma contamination by 4, 6-diamidino-2-phenylindole (DAPI) staining.

## Syngeneic tumor studies

In vivo tumor studies were performed as follows: 6-wk-old WT control and Meflin-KO female mice were inoculated subcutaneously in their right flanks with $1.0 \times 10^6$ MC-38 cells suspended in 100 $\mu$l of PBS or orthotopically in their fourth right mammary fat pads with $5.0 \times 10^5$ EO771 cells suspended in 50 $\mu$l of PBS. The volumes of MC-38 tumors were measured and calculated two to three times per week using the modified ellipsoid formula: 1/2 × (length × width$^2$). Mice with tumor volumes >2,000 mm$^3$ were euthanized. Animals whose tumors were ulcerated with bleeding before progression were terminated and included in the study.

## In vivo antibody treatment

To investigate the efficacy of ICB therapy in the MC-38 subcutaneous tumor model, anti–mPD-1 (RMP1-14, RRID: AB_2800578; BioLegend) and isotype control (RTK2758; BioLegend) antibodies were administered intraperitoneally to mice on day 4, which was 4 d after tumor inoculation at a dose of 200 $\mu$g/body, followed by subsequent antibody administration on day 7 and 10 at the same dose. For EO771 orthotopic tumor models, antibody administration was initiated on day 6 after palpable tumors had formed, followed by antibody administration on day 9 and 12 at the same dose via the same route.

## Tumor growth/tumor volume analysis

A linear mixed-effects model was used to examine repeated measurement data to investigate the effect of genotype (G: WT or Meflin-KO), treatment (T: control or PD-1), and interaction between G and T on tumor volume over time. The following model was used in the analysis:

$$\log(TV) = \beta_0 + b_0 + (\beta_1 + b_1) \times days + \beta_2 \times G + \beta_3 \times T + \beta_{12} \times days \times G + \beta_{13} \times days \times T + \beta_{123} \times days \times G \times T$$

$(b_0, b_1) \sim Normal(0, \Phi)$, $\Phi = (\sigma_{B0}^2, \sigma_{B01}, \sigma_{B1}^2)$.

Here, $\beta_0, \beta_1, \beta_2, \beta_3, \beta_{12}, \beta_{13}$, and $\beta_{123}$ are the coefficients of the fixed effects. $b_0$ is the random effect of the intercept, and $b_1$ is the random effect of the slope. $\sigma_{B0}^2$ is the variance of the individual difference at the baseline, $\sigma_{B1}^2$ is the variance of the individual difference of the slope, and $\sigma_{B01}$ is the covariance of the individual difference of the baseline and the individual difference of the slope. Regression lines were used to fit a linear profile to the time courses of logarithm-transformed tumor volumes in each group. Fitting was performed using customized functions in R v.3.6.3, which integrates software from open-source packages, including lme4 and lmerTest (Bates et al, 2015; Kuznetsova et al, 2017). Visualization of the growth curves was performed using GraphPad Prism 6.

For EO771 orthotopic tumor models, we measured and analyzed tumor volumes on the day of termination. Because the data showed non-normal distribution and heteroscedasticity, we chose a non-parametric Brunner–Munzel test with permutation to analyze tumor volumes in each group. The effect size was calculated using Cliff's delta statistic method and visualized using functions in R v.3.6.3, which integrates software from an open-source package, including dabestr (Ho et al, 2019).

## Tumor processing

To isolate cells from tumors for FCM analysis, tumors were mechanically dissociated, followed by filtering through 100 and 40 $\mu$m cell strainers and centrifugation to collect the cells. For qPCR, total RNA was extracted from tumors using TRI-Reagent (Molecular Research Center), following the manufacturer's protocol. To isolate CAFs from tumors, tumors were mechanically minced and digested using the tumor dissociation kit (130-096-730; Miltenyi Biotec) in gentleMACS C-Tubes (130-093-237; Miltenyi Biotec) according to the manufacturer's instructions. Tumor samples were minced and incubated in digestion media at 37°C for 30 min in a gentleMACS Octo dissociator (130-096-427; Miltenyi Biotec). After the digestion period, cells were suspended in a cold FACS buffer (0.5% BSA and 2 mM EDTA in PBS), filtered through 70-$\mu$m filters, and centrifuged to collect the cells.

## Flow cytometry analysis

FCM staining and analysis were performed using conventional procedures. Cells were washed using a FACS buffer (0.5% BSA and 2 mM EDTA in PBS) and stained with cell–surface antibodies and Fixable Viability Dye eFlour 506 (eBioscience). For the intracellular staining of Foxp3 and Ki-67, cells were fixed and permeabilized using the Foxp3/Transcription Factor Staining Buffer

Set (eBioscience) according to the manufacturer's instructions, followed by staining with monoclonal antibodies against Foxp3 (1:50 dilution) and Ki-67 (1:100 dilution). After washing, cells were analyzed with a BD LSRFortessa X-20 flow cytometer (BD Biosciences) and FlowJo (TreeStar) software. In this study, the following anti-mouse antibodies labelled with fluorescent dyes were used: CD3-Alexa Fluor (AF) 700 (clone 17A2, RRID: AB_493697), CD4-APC-Fire 750 (clone RM4-4, RRID: AB_2715955), CD8a-Brilliant Violet (BV) 785 (clone 53-6.7, RRID: AB_2562610), CD25-BV605 (clone PC61, RRID: AB_2563059), PD-1-BV421 (clone 29F.1A12, RRID: AB_2561447), PD-L1-APC (clone 10F.9G2, RRID: AB_10612741), CD137-PE (clone 17B5, RRID: AB_2205693), Tim-3-PE-Cyanine7 (clone RMT3-23, RRID: AB_2571932), TIGIT-BV421 (clone 1G9, RRID: AB_2687311), CD96-PE (clone 3.3, RRID: AB_1279389), CD226-BV711 (clone TX42.1, RRID: AB_2715922) (all from BioLegend); FoxP3-PerCP-Cyanine5.5 (clone FJK-16s, RRID: AB_914351), Ki-67-FITC (clone SolA15, RRID: AB_11151689), ICOS-APC (clone 7E.17G9, RRID: AB_2716947) (all from eBioscience); LAG-3-BV711 (clone C9B7W, Cat. no. 563179; BD Bioscience); and 2B4-AF488 (clone 244F4, Cat. no. NBP2-00223AF488; Novus Biologicals).

## qPCR

Total RNA was purified from whole tumors and cultured CAFs using the RNeasy Mini Kit (Cat. no. 74104; QIAGEN) according to the manufacturer's instructions. Purified RNA samples were reverse-transcribed using ReverTra Ace (Cat. no. TRT-101; Toyobo) with oligo dT and random primers. qPCR of the generated cDNAs was performed with TaqMan Gene Expression Master Mix (Cat. no. 4369016; Applied Biosystems) using a StepOnePlus thermal cycler (Applied Biosystems). Applied Biosystems synthesizes customized TaqMan probes and primers for the mouse Meflin (*Islr*) coding sequence. Cycling conditions were as follows: 50°C for 2 min, 95°C for 10 min, 40 cycles of 95°C for 15 s, and 60°C for 1 min. The data were analyzed using the $2^{-\Delta\Delta Ct}$ method and normalized to *Col1a1* (Mm00801666_g1).

## RNA in situ hybridization

To detect single mRNA molecules, an RNA in situ hybridization (ISH) assay based on RNAscope technology (Advanced Cell Diagnostics, ACD) was performed on formalin-fixed and paraffin-embedded (FFPE) human and mouse tissue samples and fixed cultured cells. Sample preparation was performed according to the manufacturer's protocol for RNAscope on tissue samples (ACD, document number 322452) or the technical note for fixed cultured cells (ACD, document number MK-50010). ISH was performed according to the protocol of the RNAscope 2.5 HD Detection Reagent-BROWN (ACD, Cat. no. 322310), the 2.5 HD Duplex Detection Kit (ACD, Cat. no. 322500), or Multiplex Fluorescent Reagent Kit v2 (ACD, Cat. no. 323100). Briefly, FFPE sections were deparaffinized in xylene and 100% ethanol and dried completely for 5 min at room temperature (RT). After incubation with $H_2O_2$ solution for 10 min at RT, slides were treated in a boiling target retrieval solution (>98°C) for 15 min, washed in distilled water, dehydrated in 100% ethanol, and dried completely. Finally, the slides were incubated with Protease Plus (ACD) at 40°C for 30 min. After washing in distilled water, slides were incubated with the relevant probes at 40°C for 2 h, followed by amplification through sequential amplification using AMP-1 to AMP-6

reagents (ACD). Staining was visualized with 3,3-diaminobenzidine (DAB) followed by counterstaining with hematoxylin. To detect the expression of more than two mRNAs on the same slides using bright field microscopy, we used the 2.5 HD Duplex Detection Kit (Fast Green and Fast Red, ACD), staining with alkaline phosphatase (ALP), and DAB followed by counterstaining with hematoxylin. To detect the expression of more than two mRNAs on the same slides by fluorescent microscopy, we used Multiplex Fluorescent Reagent Kit v2 (ACD), staining with fluorophores TSA (tyramide signal amplification)- cyanine 3 and - cyanine 5 (AKOYA Bioscience), followed by counterstaining with DAPI. In this study, the following six different probes against genes of interest were used: Hs-*ISLR* (human Meflin) (NM_005545.3, region 275–1322; Cat. no. 455481), Mm-*Islr* (mouse Meflin) (NM_012043.4, region 763–1690; Cat. no. 450041), Hs-*ACTA2* (NM_001613.2, region 10–1341; Cat. no. 311811), Hs-*PDGFRA* (NM_006206.4, region 844–1774; Cat. no. 604481), and Hs-*PDPN* (NM_001006625.1, region 911–2045; Cat. no. 539751).

## Assessment of ISH staining

To assess Meflin expression in the stroma, we first counted total stromal cells other than immune cells including macrophages identified by their morphology. The Meflin-positive cells were then counted and divided by the total stromal cells to calculate the percentage in the stroma. When cells had four and more dots or had any clusters of ISH signals, we considered them positive. We semi-quantitatively scored the expression of Meflin in each specimen (Fig 1C) or each patient (Figs 3B and 4B) according to the percentage of Meflin-positive cells. Specifically, 0% and 1–5% stromal cell expressing Meflin were combined into the score "<5" which referred to "0" in Fig 1C; thereafter, we scored 5–10% as "5," 10–15% as "10," 15–20% as "15" and so forth. For assessing KP mice specimens (Fig 1D), quantitative values were used except for INV to clarify the difference between TA and PIL.

## IHC assay using the Opal-IHC kit and immunofluorescent staining combined with RNAscope

For IHC, FFPE tissue sections were deparaffinized and subjected to antigen retrieval using a target retrieval solution (Dako or Novocastra) at pH 6, 7, or 9 for 30 min in an electric kettle, followed by IHC using conventional procedures. For multiplex immunofluorescent (IF) staining of TILs, the Opal 7 Tumor Infiltrating Lymphocyte Kit (Cat. no. OP7TL3001KT; Akoya Bioscience) was used to stain CD4, CD8, CD20, FoxP3, CD45RO, and pan-cytokeratin, according to the manufacturer's instructions. The sections were mounted using PermaFluor Aqueous Mounting Medium (Cat. no. TA-030-FM; Thermo Fisher Scientific), followed by scanning using a Vectra slide scanner (Akoya Bioscience). Five randomly selected multispectral high-powered field images of each section were captured using an automated imaging system (Vectra ver. 3.0, Akoya Bioscience) and loaded into user-trainable image analysis software (InForm, Akoya Bioscience), which allows the automatic recognition of tissue regions and individual cells to perform cell classification and phenotyping.

For the combined detection of mRNAs by ISH and proteins by IF, we first stained FFPE sections by ISH with the fluorophore TSA-Cyanine 5 for visualization. The ISH-stained slides were washed

with PBS twice and incubated with blocking buffer containing 10% serum (of the host from which secondary antibodies were derived) for 30 min at RT, followed by incubation with primary antibodies diluted in PBS at 4°C overnight. After washing with PBS, the slides were incubated with Alexa Fluor 488/594-conjugated secondary antibodies (Thermo Fisher Scientific) for 30 min at RT, followed by incubation with DAPI for 30 s at RT and mounting with PermaFluor aqueous mounting medium.

For proteins for which corresponding antibodies that work for IF after performing ISH are unavailable, such as LCA and podoplanin (Fig S2A and B), we first performed IF using Opal fluorophores (NEL810001KT; Akoya Biosciences) according to the manufacturer's instructions. Briefly, FFPE sections were deparaffinized, followed by antigen retrieval in AR6 or AR9 (AR600250ML, AR900250ML; Akoya Biosciences) for 15 min at 98°C. After blocking with Antibody Diluent/Blocking (ARD1001EA; Akoya Biosciences), the sections were incubated with primary antibodies for 2 h at RT or overnight at 4°C, washed three times in Tris-buffered saline (150 mM NaCl and 25 mM Tris–HCl) with 0.05% polyoxyethylene (20) sorbitan monolaurate (Tween 20) (TBST), and incubated in Opal Polymer HRP secondary antibody (ARH1001EA; Akoya Biosciences) for 10 min at RT. After three washes in TBST, Opal 570 in 1X Plus Amplification Diluent (1: 125) was added and reacted for 10 min at RT. After three washes in TBST, we performed ISH using RNAscope Multiplex Fluorescent Reagent Kit v2 (323100; ACD) according to the manufacturer's instructions. For mRNA retrieval and the removal of antibody-HRP complexes, the slides were immersed in 1X Target Retrieval Buffer using a pressure cooker (Cat. no. SR-MP300; Panasonic) for 15 min, followed by washing the sections in distilled water and dehydrating them in 100% ethanol. After drying the slides for 5 min at 60°C, they were treated with Proteinase Plus (Cat. no. 322331; ACD) for 30 min at 40°C. The sections were washed once in deionized water, incubated with target probes for 2 h at 40°C, washed twice in 1× wash buffer, and then incubated in amplification reagents (AMP1-3), followed by HRP-C1 Reagent (Cat. no. 323104; ACD). The signals were amplified with a TSA-Plus Cyanine 5 System (Cat. no. NEL745001KT; PerkinElmer), followed by mounting the sections with PermaFluor Aqueous Mounting Medium. Fluorescence was examined using an inverse immunofluorescence microscope BZ-X710 (Keyence) with optical sectioning.

The following antibodies were used in the present study: mouse monoclonal anti-E-cadherin (clone 36, Cat. no. 610181, dilution 1:500; BD Biosciences), mouse monoclonal anti-human CD31 (clone JC70A, Cat. no. M0823, dilution 1:250; Dako), mouse monoclonal anti-LCA (clone 2B11 + PD7/26, Cat. no. IR751; Dako), rabbit polyclonal anti-Col1a1 (Cat. no. NB600-408, dilution 1:500; Novus), mouse monoclonal anti-$\alpha$-SMA (clone 1A4, Cat. no. M0851, dilution 1:500; Dako), mouse monoclonal anti-Podoplanin (clone D2-40, Cat. no. ab77854, dilution 1:100; Abcam), and rat monoclonal anti-mouse CD31 (clone SZ31, Cat. no. DIA-310, dilution 1:100; Dianova).

### Statistical analysis

We used GraphPad Prism 6 or R v.4.1.2 for statistical analysis. Patient characteristics and binary outcomes were compared between the two groups using Fisher's exact test or the Mann–Whitney U test. Survival was analyzed using the Kaplan–Meier approach and the log-rank Mantel–Cox test, as well as the Cox proportional hazards regression model. Variables with $P$-values < 0.2 on the univariate Cox models were included in the multivariate analyses. Also, Meflin expression status, PD-L1 TPS, and the variables with $P$-values < 0.1 on the analyses of patient characteristics were included in the final model, irrespective of their statistical significance. The magnitudes of the associations were summarized using hazard ratios with 95% confidence intervals (CIs). To evaluate the predictive value for responders, a receiver operating characteristic curve (ROC) for discrete variables was created by plotting the true-positive rate against the false-positive rate at each threshold setting. The area under the curve (AUC) shown in the plot summarizes the performance of discrete variables. The cut-off value of discrete variables in which the sum of sensitivity and specificity was the maximum was detected. For murine experiments, all data are representative of at least two to three independent experiments with three to six mice in each in vivo experiment. The data were expressed as means with 95% CIs unless otherwise specified. The relationships between groups were compared using a two-tailed unpaired $t$ test with Welch's correction unless otherwise specified. For multiple testing, the Holm–Bonferroni method was employed. Survival was analyzed using the Kaplan–Meier approach and the log-rank Mantel–Cox test. Statistical significance was set at $P < 0.05$.

## Data Availability

This manuscript does not have large-scale data sets to deposit to the public databases.

## Supplementary Information

## Acknowledgements

We thank Kaori Ushida and Kozo Uchiyama (Nagoya University) for their support in immunostaining and tissue preparation and the staff of the Division of Experimental Animals (Nagoya University) for their support in animal experiments. We thank Editage (http://www.editage.com) for editing and reviewing this manuscript for English language. This work was supported by a Grant-in-Aid for Scientific Research (B) (18H02638 to A Enomoto, 20H03467 to M Takahashi, 20H03528 to Y Ando) and a Grant-in-Aid for Research Activity Start-up (20K22807 to Y Miyai) commissioned by Japan Society for the Promotion of Science; Nagoya University Hospital Funding for Clinical Research (to A Enomoto); AMED-CREST (Japan Agency for Medical Research and Development, Core Research for Evolutional Science and Technology; 20gm0810007h0105 and 20gm1210009s0102 to A Enomoto), the Project for Cancer Research and Therapeutic Evolution (P-CREATE) from AMED (20cm0106377h0001 to A Enomoto), and Aichi Cancer Research Foundation (to Y Miyai).

### Author Contributions

Y Miyai: conceptualization, data curation, formal analysis, funding acquisition, validation, investigation, visualization, methodology, and writing—original draft, review, and editing.

D Sugiyama: data curation, formal analysis, and methodology.
T Hase: resources.
N Asai: resources.
T Taki: resources and investigation.
K Nishida: formal analysis.
T Fukui: resources.
TF Chen-Yoshikawa: resources.
H Kobayashi: resources.
S Mii: resources, investigation, and writing—review and editing.
Y Shiraki: resources and investigation.
Y Hasegawa: resources.
H Nishikawa: resources and supervision.
Y Ando: resources, supervision, and funding acquisition.
M Takahashi: resources, supervision, and funding acquisition.
A Enomoto: supervision, funding acquisition, project administration, and writing—original draft, review, and editing.

## Conflict of Interest Statement

Y Miyai is the primary inventor on a pending patent PCT/JP2019/004521 related to the current work belonging to Nagoya University. T Hase has received honoraria from AstraZeneca, Ono Pharmaceutical, Chugai Pharmaceutical, and Bristol-Myers Squibb and received research funding from Boehringer Ingelheim and Taiho Pharmaceutical outside the scope of this work. Y Hasegawa has received honoraria from AstraZeneca, Ono Pharmaceutical, Chugai Pharmaceutical, and Taiho Pharmaceutical outside the scope of this work. H Nishikawa received honoraria and research funding from Ono Pharmaceutical, Chugai Pharmaceutical, MSD, and Bristol-Myers Squibb, and research funding from Taiho Pharmaceutical, Daiichi-Sankyo, Kyowa Kirin, Zenyaku Kogyo, Oncolys BioPharma, Debiopharma, Asahi-Kasei, Sysmex, Fujifilm, SRL, Astellas Pharmaceutical, Sumitomo Dainippon Pharma, and BD Japan outside of this study. Y A has received honoraria from Chugai Pharmaceutical and received research funding from Novartis, Ono Pharmaceutical, Chugai Pharmaceutical, Bayer Yakuhin, and Yakult Honsha outside the scope of this work. The other authors declare no competing interests.

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
