## [Reviewer comments · Life Science Alliance]

Life Science Alliance

Meflin-positive cancer-associated fibroblasts enhance tumor response to immune checkpoint blockade

Yuki Miyai, Daisuke Sugiyama, Tetsunari Hase, Naoya Asai, Tetsuro Taki, Kazuki Nishida, Takayuki Fukui, Toyofumi Fengshi Chen-Yoshikawa, Hiroki Kobayashi, Shinji Mii, Yukihiro Shiraki, YOSHINORI HASEGAWA, Hiroyoshi Nishikawa, Yuichi Ando, Masahide Takahashi, and Atsushi Enomoto

DOI: <https://doi.org/10.26508/lsa.202101230>

Corresponding author(s): *Atsushi Enomoto, Nagoya University*

Review Timeline:

Submission Date:	2021-09-08
Editorial Decision:	2021-10-27
Revision Received:	2022-01-21
Editorial Decision:	2022-02-17
Revision Received:	2022-02-18
Accepted:	2022-02-18

Transaction Report:

October 27, 2021

Re: Life Science Alliance manuscript #LSA-2021-01230-T

Atsushi Enomoto
Nagoya University
Institute for Advanced Research
Furo-cho, Chikusa-ku
Nagoya, Aichi
Japan

Dear Dr. Enomoto,

Thank you for submitting your manuscript entitled "Meflin-positive cancer-associated fibroblasts enhance tumor response to immune checkpoint blockade" to Life Science Alliance. The manuscript was assessed by expert reviewers, whose comments are appended to this letter. We invite you to submit a revised manuscript addressing the Reviewer comments.

Thank you for this interesting contribution to Life Science Alliance. We are looking forward to receiving your revised manuscript.

Sincerely,

B. MANUSCRIPT ORGANIZATION AND FORMATTING:

Reviewer #2 (Comments to the Authors (Required)):

In their manuscript Miyai et al describe a role for Meflin+ CAFs in response to immune checkpoint blockade. This group has previously shown that Meflin is expressed in CAFs in pancreatic and colorectal cancer. Here they show that Meflin is expressed in lung cancer associated CAFs in patients and in mouse models. They further show that a high proportion of Meflin+ CAFs correlates with a favorable response to immune checkpoint blockade (ICB) therapy in patients with NSCLC. They could not achieve response of mice with lung cancer to ICB, so they used alternative tumor models (MC38 colon cancer cells and E0771 breast cancer cells) to show that loss of Meflin reduced the response to immunotherapy. Together they propose that Meflin marks a tumor repressive subpopulation of CAFs.

This is an interesting and thorough study, describing an important aspect of CAF biology. There are several issues that must be clarified or addressed for this manuscript to be accepted for publication.

1. The authors claim in page 6 that Meflin is expressed in SMA low/neg CAFs. Yet in page 5 they state that Meflin expression was observed "in cells positive for CAF markers, such as collagen type I alpha 1 (COL1A1), α -SMA, and podoplanin (PDPN)". How do these statements coincide? It is possible that the different methods used (RNA-FISH vs protein IHC) lead to somewhat different results. The authors should do a more extensive analysis to support the claim that Meflin is expressed in SMA low/neg CAFs. Is it iCAFs? apCAFs? They should test for overlap with other CAF markers, and perhaps try also markers for antigen-presenting CAFs. Otherwise, the claim that Meflin marks a distinct subset of CAF should be toned down.
2. In Fig 3, the author claim that Meflin+ CAF levels correlate with disease outcome. But what about total CAF levels? They should quantify the total CAFs to show that it is Meflin rather than global stromal levels.
3. In Fig 6A, the authors show that the expression of some immune checkpoint molecules changed in Meflin KO mice. They checked for many factors, and therefore they must correct for multiple comparisons, to see if these changes still hold statistically.

Minor:

1. Fig 2C can be moved to supplementary
2. The discussion of fibrosis is very elaborate, whereas the authors did not monitor fibrosis (by measuring ECM, for example). This section should be shortened, unless the authors show more evidence to support the fibrosis claim.

Reviewer #3 (Comments to the Authors (Required)):

Meflin-positive cancer-associated fibroblasts enhance tumor response to immune checkpoint blockade. Miyai...Enomoto et al

This Ms focusses on CAFs in the tumor microenvironment (TME) and aims to explore whether subsets of CAFs contribute to failure or success of immune checkpoint blockade (ICB). The authors have previously described a CAF subset defined by expression of a particular GPI-linked surface protein known as meflin. This subset is found in pancreatic and colon cancers. Meflin binds BMP7, which inhibits TGF- effects including induction of fibrosis. The authors report that meflin+ CAFs correlate with higher levels of CD4+T cells, vascularity and with favorable response to ICB in non-small-cell lung cancer (NSCLC) both in patients and in mouse models.

Furthermore they present data that knocking out meflin reduces the efficacy of anti-PD-1 treatment in syngeneic murine tumor models, whereas overexpression of meflin enhances the immunotherapeutic response.

The paper addresses timely questions about ways to enhance checkpoint blockade immunotherapy dependent on a subset of CAFs that appear to enhance ICB. Overall, the extensive experimental data presented do support the conclusions of the paper, which are relevant and potentially suitable for publication. However, some of the data in the figures is not presented in a totally convincing fashion; despite the extensive supplementary methods section (and rather sketchy methods in the main text) if one reads the main text and figures as most readers will do, they do not adequately support the authors claims. This is because the evidence for the association of meflin+ CAFs with better responses to ICB is presented in a way that is not clear from the figures. These data rely on detection methods that are key to the whole story but need to be presented in a way that is more convincing -

including higher magnifications, better explanation of the criteria and analysis of the results. These deficits should be fixable by a significant revision of the data presentation. I understand the challenges of scoring and presenting these data but I believe that a better presentation is required to provide a convincing description for the readers.

Fig. 1A,B. The meflin ISH is not compelling as presented- it is not clear how the distinction is made between meflin+ CAFs (black arrows) and meflin- macrophages (white arrows) - in most panels other than INV they do not look very different to me and no clear criteria are given for the identification of each cell type (CAF vs macrophage. Similarly, in panel 1E it is essentially impossible to distinguish the cells marked in green for CAF markers from others not marked in green and one cannot even really see the cells marked yellow, let alone be convinced that they are double-labelled. No details are given in the main text or figures of IHC or the ISH methods or of the scoring; e.g., were the data in C and D scored blindly? Are the differences among the histograms statistically different? Fig 1F is too small to see whether or not there is any co-expression of meflin and SM-A. Even the detailed supplementary methods do not address all of these concerns. Similar concerns apply to much of Fig. 2. Since the scoring of meflin+ CAFs is the basis for the rest of the paper, all these characterizations need to be much improved to be convincing - higher magnification figures, better description of methods and statistics. Without such improvements, one has to take the authors claims later in the paper (Figs 2, 3, 4) on faith - that is not how it should be; the authors need to show the reader data that convinces them of the validity of the claims.

Page 5. few readers will know what lepidic means - I did not.

Figs. 4 and EV4. Again, were these scored blindly? It is hard to tell from the IHC figures how clearly the different immune cell types can be distinguished.

I could not properly evaluate the statistical tests of the data used in Fig. 5 so you need to consult a referee that can. Assuming the statistics are OK, then the figure does demonstrate a role for meflin in the immunosuppression effects and the data on the syngeneic murine models (Figs 6 and 7) further support a role for meflin in creating a better tumor microenvironment for anti-PD-1 immunotherapy.

'Referee Cross-Comments'

I agree with the comments of Reviewer #2 - they address different weaknesses than my review. The authors need to address both.

January 21, 2022

Dr. Eric Sawey,
Executive Editor
Life Science Alliance

Dear Dr. Eric Sawey,

Thank you so much for reviewing our manuscript (manuscript ID: LSA-2021-01230-T) and providing us with the opportunity to revise our manuscript. We are grateful to you and the reviewers for the critical and suggestive comments on the original version of our manuscript. We carefully examined the reviewer's comments and performed additional analysis to try to address all the reviewer's comments, as indicated on the attached pages. We included the additional or modified data in Figures 1E, 2, 4C and Supplementary Figure S2C and D and made some modifications in the Result, Discussion, Materials and Methods, and Reference sections. In the revised manuscript, we highlighted major changes made from the original version in yellow.

We hope that the revised version of our paper is now suitable for publication in Life Science Alliance.

Yours sincerely,

Atsushi Enomoto, M.D., Ph.D.
Associate Professor, Department of Pathology
Nagoya University Graduate School of Medicine
65 Tsurumai-cho, Showa-ku
Nagoya 466-8550, Japan
Tel: 81-52-744-2093
Fax: 81-52-744-2098
E-mail: enomoto@iar.nagoya-u.ac.jp

Response to Reviewers

Response to Reviewer #2:

In their manuscript Miyai et al describe a role for Meflin+ CAFs in response to immune checkpoint blockade. This group has previously shown that Meflin is expressed in CAFs in pancreatic and colorectal cancer. Here they show that Meflin is expressed in lung cancer associated CAFs in patients and in mouse models. They further show that a high proportion of Meflin+ CAFs correlates with a favorable response to immune checkpoint blockade (ICB) therapy in patients with NSCLC. They could not achieve response of mice with lung cancer to ICB, so they used alternative tumor models (MC38 colon cancer cells and E0771 breast cancer cells) to show that loss of Meflin reduced the response to immunotherapy. Together they propose that Meflin marks a tumor repressive subpopulation of CAFs. This is an interesting and thorough study, describing an important aspect of CAF biology. There are several issues that must be clarified or addressed for this manuscript to be accepted for publication.

Response: Thank you for your positive comments on our manuscript. We are grateful to the reviewer for critical comments and useful suggestions that have helped us improve our manuscript. As indicated in our responses below, we have taken all of your comments and suggestions into account in the revised version of our manuscript. We included the additional data in Figures 1E, 2, 4C and Supplementary Figure S2C and D, and made some modifications in the Result, Discussion, and Materials and Methods sections. In the revised manuscript, we highlighted major changes made from the original version in yellow.

1) The authors claim in page 6 that Meflin is expressed in SMA low/neg CAFs. Yet in page 5 they state that Meflin expression was observed "in cells positive for CAF markers, such as collagen type I alpha 1 (COL1A1), α -SMA, and podoplanin (PDPN)". How do these statements coincide? It is possible that the different methods used (RNA-FISH vs protein IHC) lead to somewhat different results. The authors should do a more extensive analysis to support the claim that Meflin is expressed in SMA low/neg CAFs. Is it iCAFs? apCAFs? They should test for overlap with other CAF markers, and perhaps try also markers for antigen-presenting CAFs. Otherwise, the claim that Meflin marks a distinct subset of CAF should be toned down.

Thank you for the comment. Unfortunately, there are no commercially available antibodies that work well for Meflin immunostainings; therefore, we adopted *in situ* hybridization (ISH)

as well as used single-cell RNA sequencing datasets for the analysis and quantification of Meflin expression throughout the study. The combination of immunofluorescent (IF) staining and ISH showed that approximately 30% of Meflin mRNA-positive cells were positive for α -SMA protein (**Supplementary Fig S2B** in the revised manuscript). The data seemed to contradict with the results of 2-color duplex ISH shown in **Fig 1F**, which showed that only a minor population (~10%) of CAFs was double-positive for *ACTA2* and *ISLR*, which encodes α -SMA and Meflin, respectively. Given these data, we agree with the reviewer's suggestion that different methods may lead to different results. We speculate that the discrepancy may be attributed to CAF diversities among patients, differences in half-life between α -SMA mRNA and protein, or differences in detection sensitivity between IF and ISH.

In the revised manuscript, we attempted to show that Meflin (*ISLR*) is expressed in CAFs that are weakly positive or negative for *ACTA2* (termed *ACTA2* low/neg CAFs) and examined whether Meflin-positive CAFs are overlapped with iCAF or apCAF by analyzing the single-cell RNA sequencing dataset deposited by Lambrechts et al. (Nat Med, 24:1277-1289, 2018) (**Fig 2** in the revised manuscript). The result showed that Meflin (*ISLR*) expression was significantly enriched in *ACTA2* low/neg, *IL6* low/neg, and *HLA-DRA* low/neg subsets but not in *ACTA2* high (myCAFs), *IL6* high (iCAFs), and *HLA-DRA* high (apCAFs) subsets (**Fig 2** in the revised manuscript). We believe that the data clearly showed that Meflin-positive cells represent a CAF subset that is distinct from myCAF, iCAF and apCAF. In the revised manuscript, we described the finding in the text as follows:

Page 6, line 12: “Further analysis focusing on the fibroblast cluster of the single-cell RNA sequencing data (Lambrecht et al, 2018) confirmed that Meflin expression was enriched in *ACTA2*^{low/neg}, *IL6*^{low/neg}, or *HLA-DRA*^{low/neg} subsets, indicating that Meflin⁺ CAFs represent a CAF subset distinct from myCAF, iCAF, and apCAF (**Fig 2**).”

2) In Fig 3, the author claim that Meflin+ CAF levels correlate with disease outcome. But what about total CAF levels? They should quantify the total CAFs to show that it is Meflin rather than global stromal levels.

Response: We appreciate your important comment. In the revised manuscript, we counted the total numbers of fibroblasts found in high-power fields of tissue sections obtained from patients with non-small cell lung carcinoma (NSCLC) who underwent surgery (n = 38), followed by quantification (**Fig 4C** in the revised manuscript). The data showed there was no significant difference in the numbers of total fibroblasts between the Meflin-high and -low groups, implying that the number of Meflin-positive CAFs, but not total fibroblasts, is vital for determining the outcome of the patients with NSCLC treated with immune checkpoint

blockade. We thank the reviewer's comment that improve our manuscript. In the revised manuscript, we presented the data in **Fig. 4C** and edited the text as follows:

Page 7, Lines 20: “We also evaluated the average total numbers of fibroblasts based on cell morphology, and found that they were comparable between the Meflin-high and -low groups (**Fig 4C**).”

3) In Fig 6A, the authors show that the expression of some immune checkpoint molecules changed in Meflin KO mice. They checked for many factors, and therefore they must correct for multiple comparisons, to see if these changes still hold statistically.

Response: We apologize for not providing clear descriptions in the original version of our manuscript. We would like to point out that the experiment shown in **Fig 7** in the revised manuscript (**Fig 6** in the original version of our manuscript) was an exploratory analysis, in which we analyzed six sets of pooled cells isolated from independent tumors developed in three mice for each set (eighteen wild-type and eighteen Meflin knock out mice in total) and confirmed the results by performing two independent experiments. To the best of our understanding, it would be difficult to adopt correction methods for multiple comparisons to analyze the data obtained from exploratory experiments. In the revised manuscript, we just mentioned that the data shown in Fig 7 are based on exploratory analyses as follows:

Page 11, Line 11: " These exploratory analyses suggest that Meflin expression in CAFs is associated with TIL activation in mice, but not their recruitment or infiltration into tumors."

We would welcome any further advice and suggestions by the reviewer regarding this issue.

Minor:

1) Fig 2C can be moved to supplementary

Response: We moved **Fig 2C** in the original manuscript to **Supplementary Fig S3** in the revised manuscript.

2) The discussion of fibrosis is very elaborate, whereas the authors did not monitor fibrosis (by measuring ECM, for example). This section should be shortened, unless the authors show more evidence to support the fibrosis claim.

Response: Thank you for the advice. We shortened that section by removing sentences that were not based on the experiments in the current study as follows:

The removed sentences from the original version of the manuscript: " Consistent with this, the induction of cardiac fibrosis and pancreatic carcinogenesis in Meflin-KO mice resulted in enhanced fibrosis and infiltration of α -SMA⁺ myofibroblasts or CAFs compared to WT mice (Mizutani et al, 2019; Hara et al, 2019). Interestingly, Meflin-KO hearts exhibit an increased stiffness compared to WT hearts in a cardiac fibrosis model, leading to the hypothesis that the fundamental function of Meflin is to inhibit tissue fibrosis and interstitial pressure elevation in chronic disease conditions (Hara et al, 2019)."

Response to Reviewer #3:

This Ms focusses on CAFs in the tumor microenvironment (TME) and aims to explore whether subsets of CAFs contribute to failure or success of immune checkpoint blockade (ICB). The authors have previously described a CAF subset defined by expression of a particular GPI-linked surface protein known as meflin. This subset is found in pancreatic and colon cancers. Meflin binds BMP7, which inhibits TGF- β effects including induction of fibrosis. The authors report that meflin⁺ CAFs correlate with higher levels of CD4⁺T cells, vascularity and with favorable response to ICB in non-small-cell lung cancer (NSCLC) both in patients and in mouse models. Furthermore they present data that knocking out meflin reduces the efficacy of anti-PD-1 treatment in syngeneic murine tumor models, whereas overexpression of meflin enhances the immunotherapeutic response.

The paper addresses timely questions about ways to enhance checkpoint blockade immunotherapy dependent on a subset of CAFs that appear to enhance ICB. Overall, the extensive experimental data presented do support the conclusions of the paper, which are relevant and potentially suitable for publication. However, some of the data in the figures is not presented in a totally convincing fashion; despite the extensive supplementary methods section (and rather sketchy methods in the main text) if one reads the main text and figures as most readers will do, they do not adequately support the authors claims. This is because the evidence for the association of meflin⁺ CAFs with better responses to ICB is presented in a way that is not clear from the figures. These data rely on detection methods that are key to the whole story but need to be presented in a way that is more convincing - including higher magnifications, better explanation of the criteria and analysis of the results. These deficits

should be fixable by a significant revision of the data presentation. I understand the challenges of scoring and presenting these data but I believe that a better presentation is required to provide a convincing description for the readers.

Response: Thank you for your positive comments on our manuscript. We are also grateful to the reviewer for critical comments and useful suggestions that have helped us improve our manuscript. As indicated in our responses below, we have taken all of your comments and suggestions into account in the revised version of our manuscript. We included the additional data in Figures 1E, 2, 4C and Supplementary Figure S2C and D, and made some modifications in the Result, Discussion, and Materials and Methods sections. In the revised manuscript, we highlighted major changes made from the original version in yellow.

1) Fig.1A,B. The meflin ISH is not compelling as presented- it is not clear how the distinction is made between meflin+ CAFs (black arrows) and meflin- macrophages (white arrows) - in most panels other than INV they do not look very different to me and no clear criteria are given for the identification of each cell type (CAF vs macrophage. Similarly, in panel 1E it is essentially impossible to distinguish the cells marked in green for CAF markers from others not marked in green and one cannot even really see the cells marked yellow, let alone be convinced that they are double-labelled. No details are given in the main text or figures of IHC or the ISH methods or of the scoring; e.g., were the data in C and D scored blindly? Are the differences among the histograms statistically different?

Response: We apologize for our unclear presentations of the *in situ* hybridization (ISH) data on formalin-fixed paraffin-embedded tissue sections. We used the RNAscope technology (Advanced Cell Diagnostics) to perform ISH, in which positive signals were detected as small dots in the cytoplasm or nuclei. We therefore agree with the reviewer's concern that we could not identify the morphology and types of cells that were positive for the ISH signals. We also appreciate the issue raised by the reviewer that it is difficult to discriminate fibroblasts from macrophages just based on their morphology.

Our current study is based on the premise that Meflin is not expressed in macrophages both in human and mouse tumors. We previously demonstrated that Meflin expression is quite specific to fibroblasts, and it is not expressed by other cell types including macrophages and other immune cells and endothelial, epithelial, and smooth muscle cells (Maeda et al., *Sci Rep* 2016; 6:22288). The single-cell RNA sequencing analysis presented in the present study also showed that Meflin expression was specific to fibroblasts and it was not found in myeloid cells that include macrophages (**Fig 1E** in the revised manuscript). We also showed that Meflin was not expressed in CD45-positive cells that include macrophages by

multiplex immunofluorescent staining, further supporting the notion that Meflin is not expressed in macrophages (**Supplementary Fig S2A**). We hope that the reviewer appreciate that Meflin-positive cells are different from macrophages (Meflin negative) that have phagocytosed black dusts. Taken these data together, we assume that Meflin-positive cells represent fibroblasts in the ISH analysis. It remains possible, however, that Meflin is expressed by minor populations of macrophages, which will be the subject of our study in the future.

Regarding the lack of clarity of the ISH results, we provided high magnification images of dual-color chromogenic ISH experiments on tissue sections of invasive adenocarcinoma (**Supplementary Fig S2C and D** in the revised manuscript). We hope that the reviewer and readers appreciate that each CAF is positive for green and red dots that indicate positivity for the indicated genes to various degrees. The cells that were positive for both red and green dots were counted as double-positive cells. As described in the Materials and Methods section, we considered the cells positive for the indicated gene when those had 4 and more dots or had any clusters of ISH signals (page 21, line 20 of the revised manuscript).

Regarding the scoring method, all specimens were analyzed blindly by trained pathologists including the corresponding author (A.E.). As for the reviewer's concern on statistical difference between cell fractions in the data shown in **Fig 1F**, we found that it was difficult to perform statistical analysis in this form, then we transform the graphs to be analyzed as shown in Supplementary Fig S2D in the revised manuscript.

Fig 1F is too small to see whether or not there is any co-expression of meflin and SM-A. Even the detailed supplementary methods do not address all of these concerns. Similar concerns apply to much of Fig. 2. Since the scoring of meflin+ CAFs is the basis for the rest of the paper, all these characterizations need to be much improved to be convincing - higher magnification figures, better description of methods and statistics. Without such improvements, one has to take the authors claims later in the paper (Figs 2, 3, 4) on faith - that is not how it should be; the authors need to show the reader data that convinces them of the validity of the claims.

Response: We added higher magnification images of **Fig 1E** (**Fig 1F** in the original manuscript) in **Fig 2** in the revised manuscript to help the reviewer and readers understand the distinct expression pattern of Meflin and other CAF markers in fibroblasts included in the single-cell RNA sequencing dataset. In **Fig 2**, we provided enlarged tSNE plots of the fibroblast cluster (left panels), density plots (middle panels), and violin/box/scatter plots (right panels), all of which clearly showed that Meflin-positive CAFs represent a unique CAF

subset that is different from myCAF, iCAF, and apCAF. In the revised manuscript, we described the detail of the analysis in the Materials and Method section (page 15, line 15).

Regarding the reviewer's concern on scoring methods of the signals of ISH analysis, we would appreciate it if the reviewer refers to our response described above.

2) Page 5. few readers will know what lepidic means - I did not.

We appreciate the comment. “Lepidic growth” is a pathological term in the field of lung cancer, which refers to a pattern of non-invasive cell proliferation along pre-existing alveolar walls of the lung. This growth pattern is frequently observed in lesions adjacent to invasive adenocarcinoma of the lung, as we showed in **Fig 1A** and **Supplementary Fig 1**. In the revised manuscript, we explained this as follows:

Page 5, line 10: " Interestingly, Meflin⁺ stromal cells were not observed in non-invasive tumors (adenocarcinoma in situ; AIS), whereas they were sparsely present in preinvasive lesions (PIL) with a lepidic growth, a pattern non-invasive cell proliferation along pre-existing alveolar wall, adjacent to invasive tumors (**Figs 1A and S1**)."

3) Figs. 4 and EV4. Again, were these scored blindly? It is hard to tell from the IHC figures how clearly the different immune cell types can be distinguished.

Please note that, in the revised manuscript, **Fig 4** and **EV4** in the original manuscript were moved to **Fig 5** and **Supplementary Fig S5**, respectively.

In the experiments shown in these figures, we performed 7-color multiplex staining by a Vectra slide scanner and an automated imaging system (Vectra ver 3.0, Akoya Bioscience). In that imaging, the data were obtained and analyzed by a machine learning function based on the built-in InForm software of the Vectra system. Therefore, all of the counting and quantification were performed blindly to the investigators. We described the method for this analysis in page 22, lines 1–13 in the revised manuscript. The statistical analysis was performed by the first author (Y.M.) who was not blinded to the arms we were evaluating.

4) I could not properly evaluate the statistical tests of the data used in Fig. 5 so you need to consult a referee that can. Assuming the statistics are OK, then the figure does demonstrate a role for meflin in the immunosuppression effects and the data on the syngeneic murine models (Figs 6 and 7) further support a role for meflin in creating a better tumor

microenvironment for anti-PD-1 immunotherapy.

We apologized for not showing clearly the methods for the statistical analysis shown **Fig 5** in the original version of our manuscript, which was moved to **Fig 6** in the revised manuscript.

The statistical analysis for the experiment shown in **Fig 6** was performed by by the first author (Y.M.) under the direction of one of the authors (K.N.) who is a specialist in statistics. We described the details of the methods for statistical analysis in the revised manuscript (page 18, line 4 – page 19, line 2). We believe that the method is an accurate statistical modelling for the analysis of time-dependent growth of tumors and comparison across the arms.

February 17, 2022

RE: Life Science Alliance Manuscript #LSA-2021-01230-TR

Prof. Atsushi Enomoto
Nagoya University
Department of Pathology
65 Tsurumai-cho, Showa-ku
Nagoya, Aichi 466-8550
Japan

Dear Dr. Enomoto,

Thank you for submitting your revised manuscript entitled "Meflin-positive cancer-associated fibroblasts enhance tumor response to immune checkpoint blockade". We would be happy to publish your paper in Life Science Alliance pending final revisions necessary to meet our formatting guidelines.

-please add callouts for Figures S4A-C and S6A-B to your main manuscript text

A. FINAL FILES:

B. MANUSCRIPT ORGANIZATION AND FORMATTING:

****It is Life Science Alliance policy that if requested, original data images must be made available to the editors. Failure to provide original images upon request will result in unavoidable delays in publication. Please ensure that you have access to all original**

data images prior to final submission.**

The license to publish form must be signed before your manuscript can be sent to production. A link to the electronic license to publish form will be sent to the corresponding author only. Please take a moment to check your funder requirements.

Sincerely,

Reviewer #2 (Comments to the Authors (Required)):

In their revised manuscript the authors successfully and thoroughly addressed all of my previous concerns. With regards to the statistical analysis of the data in Figure 7 I think the author's response is OK, but perhaps an expert in statistics should be consulted. As far as I am concerned this manuscript is now ready for publication in LSA.

Reviewer #3 (Comments to the Authors (Required)):

My prior description of the paper still stands and I will not repeat it. The authors have responded to each of the referees' comments in constructive fashion. The revised Ms is improved and largely supports the conclusions drawn. The data are complex and, I accept, difficult to communicate - the Ms is still a hard read even after the authors' sincere efforts to improve the figures and some of the descriptions of methods. I accept their responses and I believe the overall conclusions. It is not clear to me that further tinkering with the Ms will make it easier to digest so my recommendation is to publish it as revised, the results are intriguing and should be "out there" so that the authors and others can move on to further tests of the hypotheses presented.

February 18, 2022

RE: Life Science Alliance Manuscript #LSA-2021-01230-TRR

Prof. Atsushi Enomoto
Nagoya University
Department of Pathology
65 Tsurumai-cho, Showa-ku
Nagoya, Aichi 466-8550
Japan

Dear Dr. Enomoto,

Thank you for submitting your Research Article entitled "Meflin-positive cancer-associated fibroblasts enhance tumor response to immune checkpoint blockade". It is a pleasure to let you know that your manuscript is now accepted for publication in Life Science Alliance. Congratulations on this interesting work.

DISTRIBUTION OF MATERIALS:

Again, congratulations on a very nice paper. I hope you found the review process to be constructive and are pleased with how the manuscript was handled editorially. We look forward to future exciting submissions from your lab.

Sincerely,
